# PRIORITIZED LEVEL REPLAY

## ABSTRACT

Simulated environments with procedurally generated content have become popular benchmarks for testing systematic generalization of reinforcement learning agents. Every *level* in such an environment is algorithmically created, thereby exhibiting a unique configuration of underlying factors of variation, such as layout, positions of entities, asset appearances, or even the rules governing environment transitions. Fixed sets of training levels can be determined to aid comparison and reproducibility, and test levels can be held out to evaluate the generalization and robustness of agents. While prior work samples training levels in a direct way (e.g. uniformly) for the agent to learn from, we investigate the hypothesis that different levels provide different learning progress for an agent at specific times during training. We introduce *Prioritized Level Replay*, a general framework for estimating the future learning potential of a level given the current state of the agent's policy. We find that temporal-difference (TD) errors, while previously used to selectively sample past transitions, also prove effective for scoring a level's future learning potential when the agent *replays* (that is, revisits) that level to generate entirely new episodes of experiences from it. We report significantly improved sample-efficiency and generalization on the majority of Procgen Benchmark environments as well as two challenging MiniGrid environments. Lastly, we present a qualitative analysis showing that Prioritized Level Replay induces an implicit curriculum, taking the agent gradually from easier to harder levels.

## 1 INTRODUCTION

Environments generated using procedural content generation (PCG) have garnered increasing interest in RL research, leading to a surge of PCG environments such as MiniGrid (Chevalier-Boisvert et al., 2018), the Obstacle Tower Challenge (Juliani et al., 2019), the Procgen Benchmark (Cobbe et al., 2019), and the NetHack Learning Environment (Küttler et al., 2020). Unlike singleton environments, like those in the Arcade Learning Environment benchmark (Bellemare et al., 2013), which are exploitable by memorization and deterministic reset strategies (Ecoffet et al., 2019; 2020), PCG environments create novel environment instances or *levels* algorithmically. Every such level exhibits a unique configuration of underlying factors of variation, such as layout, positions of game entities, asset appearances, or even different rules governing environment transitions, making them a promising target for evaluating systematic generalization in RL (Risi & Togelius, 2020). Each level can be associated with a level identifier (e.g. an index, a random number generator seed, etc.) used by the PCG algorithm to generate a specific level. This allows for a clean notion of train-test split and testing on held-out levels, in line with the best practices from supervised learning.

Due to the variation among algorithmically-generated levels, a random selection of PCG levels can, in principle, correspond to levels of varying difficulty as well as reveal different—perhaps rare—environment dynamics. This diversity among levels implies that different levels hold differing learning potential for an agent at any point in training, a fact exploited by methods that both learn to generate levels as well as to solve them (Wang et al., 2019; 2020). Here, we focus on the less intrusive setting where we do not have control over level generation, but can instead *replay* (that is, revisit) any previously visited level during training to generate entirely new experiences from it.

We introduce Prioritized Level Replay, illustrated in Figure 1, a method for sampling training levels that exploits this difference among levels. Our method induces a level-sampling distribution that prioritizes levels based on the learning potential of replaying each level anew. Throughout agent training, our method updates a heuristic score appraising the agent's learning potential on a level

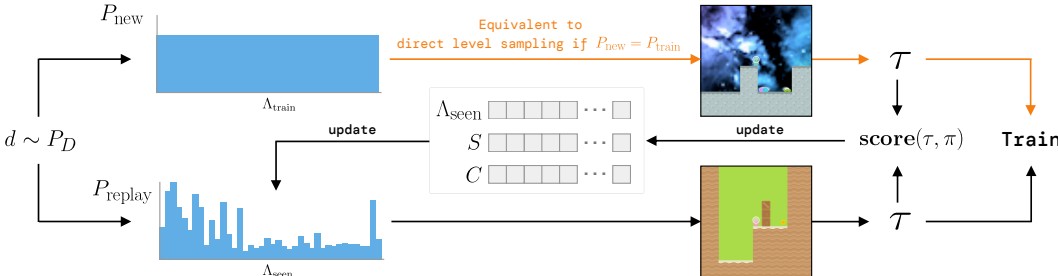

Figure 1: Overview of Prioritized Level Replay. The next level is either sampled from a distribution with support over unseen levels (top), which could be the environment's (perhaps implicit) full training-level distribution, or alternatively, sampled from the replay distribution, which prioritizes levels based on future learning potential (bottom). In either case, a trajectory $\tau$ is sampled from the next level and used to update the replay distribution. This update depends on the lists of previously seen levels $\Lambda_{\text{seen}}$, their latest estimated learning potentials $S$, and last sampled timestamps $C$.

based on temporal-difference (TD) errors collected along the last trajectory sampled from that level. Rather than sampling from the default, typically static (e.g. uniform), training level distribution, our method samples from a distribution derived from a normalization procedure over these level scores. Prioritized Level Replay does not make any assumption about how the policy is updated, and is therefore compatible with any RL method. Furthermore, our method does not rely on some external or general method for quantifying difficulty of a level, but instead derives a level score directly from the policy. The only requirements—satisfied in a wide number of settings where a simulator or game is used to collect experience—are as follows: (i) some notion of "level" exists, (ii) such levels can be sampled from the environment in an identifiable way, and (iii) given a level identifier, it is possible to set the environment to that level to be able to collect new experiences from it.

While previous works in off-policy RL devised effective methods to directly reuse *past* experiences for training (Schaul et al., 2015; Andrychowicz et al., 2017), Prioritized Level Replay uses past experiences to inform the collection of *future* experiences by assessing how much replaying each level anew will benefit learning. Our method can thus be seen as a forward-view variation of prioritized experience replay, and an online counterpart to this off-policy method for policy-gradient algorithms.

This paper makes the following core contributions: (i) we introduce a computationally efficient algorithm for adaptively prioritizing levels throughout training using a heuristic-based assessment of the learning potential of replaying each level, (ii) we empirically demonstrate our method leads to significant gains on 11 of 16 Procgen Benchmark environments and two challenging MiniGrid environments, (iii) we demonstrate our method combines with a previous leading method to set a new state-of-the-art on Procgen Benchmark, and (iv) we provide evidence that our method induces an implicit curriculum over training levels in sparse reward settings.

## 2 BACKGROUND

In this paper, we refer to a *PCG environment* as any computational process that, given a level identifier (e.g. an index or a seed), generates a *level*, defined as an environment instance exhibiting a unique configuration of its underlying factors of variation, such as layout, positions of game entities, asset appearances, or even rules that govern the environment transitions (Risi & Togelius, 2020). For example, MiniGrid's MultiRoom environment instantiates mazes with varying numbers of rooms based on the seed (Chevalier-Boisvert et al., 2018). We refer to sampling a new trajectory generated from the agent's latest policy acting on a given level $l$ as *replaying* that level $l$.

The level diversity of PCG environments makes them useful testbeds for studying the robustness and generalization ability of RL agents, measured by agent performance on unseen test levels. The standard test evaluation protocol for PCG environments consists of training the agent on a finite number of training levels, $\Lambda_{\text{train}}$ and evaluating performance on unseen test levels $\Lambda_{\text{test}}$, drawn from the set of all levels. A common variation of this protocol sets $\Lambda_{\text{train}}$ to the set of all levels, though the

agent will still effectively only sees a finite set of levels during training. Training levels are sampled from an arbitrary distribution $P_{\text{train}}(l|\Lambda_{\text{train}})$. We call this training process *direct level sampling*. In practice, for the case of a finite training set, typically $P_{\text{train}}(l|\Lambda_{\text{train}}) = \mathbf{Uniform}(l; \Lambda_{\text{train}})$. See Appendix D for the pseudocode outlining this procedure.

In generating levels of varying difficulty, PCG environments naturally lend themselves to the study of curriculum-based training protocols. For example, Justesen et al. (2018) and Wang et al. (2019; 2020) show that generating progressively harder levels improves generalization of agents across difficulty settings, demonstrating that the set of levels most useful for improving an agent's performance varies throughout the course of training. When levels cannot directly be generated, many previous works have exploited handcrafted curricula to learn policies on harder levels (Chevalier-Boisvert et al., 2018; Zhong et al., 2020). Instead, we are interested in answering how an effective curriculum over an existing set $\Lambda_{\text{train}}$ can be automatically discovered.

## 3  PRIORITIZED LEVEL REPLAY

In this section, we present Prioritized Level Replay, a level-sampling algorithm that exploits differences in the learning potential for the current agent in replaying previously visited levels. It is a drop-in replacement for the experience-collection process used in a wide range of RL algorithms. To exemplify this, we show how it is straight-forward to incorporate Prioritized Level Replay into a generic policy-gradient training loop, as described in Algorithm 1 where, for clarity, we focus on training on batches of complete trajectories (see Appendix D for detailed pseudocode on how our procedure works with $T$-step rollouts).

Our method, illustrated in Figure 1 and fully specified in Algorithm 2, induces a dynamic and non-parametric sampling distribution $P_{\text{replay}}(l|\Lambda_{\text{seen}})$ over previously visited training levels $\Lambda_{\text{seen}}$ that prioritizes visited levels with higher learning potential based on properties of the agent's past trajectories. We refer to $P_{\text{replay}}(l|\Lambda_{\text{seen}})$ as the *replay distribution*. Throughout an agent's training, our method updates this replay distribution adaptively according to a heuristic score, assigning greater weight to visited levels with higher *future* learning potential. Through variable-size lists $S$ and $C$ of equal length to $\Lambda_{\text{seen}}$, Prioritized Level Replay tracks level scores $S_i \in S$ for each visited training level $l_i$ based on the latest episode trajectory on $l_i$, as well as the episode count $C_i \in C$ at which each level $l_i \in \Lambda_{\text{seen}}$ was last sampled. Our method updates $P_{\text{replay}}$ after each terminated episode by computing a mixture of two distributions, $P_S$, based on the level scores, and $P_C$, based on how long ago each level was last sampled:

$$P_{\text{replay}} = (1 - \rho) \cdot P_S + \rho \cdot P_C, \tag{1}$$

where the staleness coefficient $\rho \in [0, 1]$ is a hyperparameter. We discuss how we compute level scores ($S_i$) parameterizing the scoring ($P_S$) distribution, and the staleness ($P_C$) distribution, in Sections 3.1 and 3.2, respectively.

Prioritized Level Replay chooses the next level at the start of every training episode by first sampling a replay-decision from a Bernoulli (or similar) distribution $P_D(d)$ to determine whether to replay a level sampled from the replay distribution $P_{\text{replay}}(l|\Lambda_{\text{seen}})$ or to experience a new, unseen level from $\Lambda_{\text{train}}$, according to some distribution $P_{\text{new}}(l|\Lambda_{\text{train}} \setminus \Lambda_{\text{seen}})$. In practice, for the case of a finite number of training levels, we implement $P_{\text{new}}$ as a uniform distribution over the remaining unseen levels. For the case of a countably infinite number of training levels, we simulate $P_{\text{new}}$ by sampling levels from $P_{\text{train}}$ until encountering an unseen level. In our experiments based on a finite number of training levels, we opt to perform a "warm start" by setting $P_D(d = 0) = 1$ (i.e. sampling a new level) as long as $|\Lambda_{\text{train}} \setminus \Lambda_{\text{seen}}| > 0$ (i.e. not every level has been seen), ensuring the replay distribution takes into account scores from at least one trajectory from each level.

The following sections describe the main steps of how Prioritized Level Replay updates the replay distribution $P_{\text{replay}}(l|\Lambda_{\text{seen}})$, namely level scoring and staleness-aware prioritization.

### 3.1  SCORING LEVELS FOR LEARNING POTENTIAL

After collecting each full trajectory $\tau$ over a completed episode on level $l_i$ using policy $\pi$, our method assigns $l_i$ a score $S_i = \mathbf{score}(\tau, \pi)$ measuring the learning potential of replaying $l_i$ in the future. We employ a function of the TD-error at timestep $t$, $\delta_t = r_t + \gamma V(s_{t+1}) - V(s_t)$, as a proxy

---

**Algorithm 1** Generic policy-gradient training loop with Prioritized Level Replay

---

**Input:** Training levels $\Lambda_{\text{train}}$ of environment $\mathcal{E}$, policy $\pi_\theta$, policy update function $\mathcal{U}(\mathcal{B}, \theta) \rightarrow \theta'$, and batch size $N_b$.

Initialize level scores $S$ and level timestamps $C$, global episode counter $c \leftarrow 0$
Initialize the ordered set of visited levels $\Lambda_{\text{seen}} = \varnothing$
Initialize experience buffer $\mathcal{B} = \varnothing$
**while** training **do**
  $\mathcal{B} \leftarrow \varnothing$
  **while** collecting experiences **do**
    $\mathcal{B} \leftarrow \mathcal{B} \cup \mathbf{collect\_experiences}(\Lambda_{\text{train}}, \Lambda_{\text{seen}}, \pi_\theta, S, C, c)$     $\triangleright$ Using Algorithm 2
  **end while**
  $\theta \leftarrow \mathcal{U}(\mathcal{B}, \theta)$     $\triangleright$ Update policy using collected experiences
**end while**

---

**Algorithm 2** Experience collection with Prioritized Level Replay

---

**Input:** Training levels $\Lambda_{\text{train}}$, visited levels $\Lambda_{\text{seen}}$, policy $\pi$, the training level distribution $P_{\text{train}}$, global level scores $S$, global level timestamps $C$, and global episode counter $c$.
**Output:** A sampled trajectory $\tau$
  $c \leftarrow c + 1$
  Sample replay decision $d \sim P_D(d)$
  **if** $d = 0$ **and** $|\Lambda_{\text{train}} \setminus \Lambda_{\text{seen}}| > 0$ **then**     $\triangleright$ Sample an unseen level, if any
    Define new index $i \leftarrow |S| + 1$
    Sample $l_i \sim P_{\text{new}}(l|\Lambda_{\text{train}}, \Lambda_{\text{seen}})$
    Add $l_i$ to $\Lambda_{\text{seen}}$, add initial value 0 to $S$ and $C$
  **else**     $\triangleright$ Sample a level for replay
    Sample $l_i \sim (1 - \rho) \cdot P_S(l|\Lambda_{\text{seen}}, S) + \rho \cdot P_C(l|\Lambda_{\text{seen}}, C, c)$
  **end if**
  Sample $\tau \sim P_\pi(\tau|l)$
  Update score $S_i \leftarrow \mathbf{score}(\tau, \pi)$ and timestamp $C_i \leftarrow c$     $\triangleright$ Update level replay lists

---

for this learning potential. Its expectation over next states is equivalent to the advantage estimate, and therefore higher-magnitude TD-errors imply greater discrepancy between expected and actual returns, making $\delta_t$ a useful measure of the learning potential in revisiting a particular state transition. To prioritize the learning potential of future experiences resulting from replaying a level, we use a scoring function based on the *average magnitude* of the Generalized Advantage Estimate (GAE; Schulman et al., 2015) over each of $T$ time steps in the latest trajectory $\tau$ from that level:

$$S_i = \mathbf{score}(\tau, \pi) = \frac{1}{T} \sum_{t=0}^{T} \left| \sum_{k=t}^{T} (\gamma\lambda)^{t-k} \delta_t \right|. \tag{2}$$

Importantly, while the GAE at time $t$ is most commonly expressed as the discounted sum of all 1-step TD-errors starting at $t$ as in Equation 2, it is equivalent to an exponentially-discounted sum of all $k$-step TD-errors from $t$, with discount factor $\lambda$. By considering all $k$-step TD-errors, the GAE mitigates the bias introduced by the bootstrap term in 1-step TD-errors. The discount factor $\lambda$ then controls the trade-off between bias and variance. Our scoring function considers the absolute value of the GAE, as we assume the degree of learning potential grows with the magnitude of the TD-error irrespective of its sign.

Another useful interpretation of Equation 2 comes from observing that the GAE at $t$ is equivalent to the L1 value loss $|\hat{V}_t - V_t|$ when applied in the context of a policy-gradient algorithm that uses GAE for its own advantage estimates—and therefore value targets $\hat{V}_t$—as done in state-of-the-art implementations of PPO (Schulman et al., 2017) used in our experiments. For this reason, we will refer to this instantiation of our algorithm as *value-based level replay*.

While we provide principled motivations for our specific choice of scoring function, we emphasize that in general, the scoring function can be any approximation of learning potential based on per-trajectory values. In Section 4, we will compare our choice of the unsigned GAE, or equivalently,

the L1 value loss, to alternative TD-error-based and uncertainty-based scoring approaches listed in Table 1.

Given level scores, the normalized outputs of a prioritization function $h$ evaluated over these scores and tuned using a temperature parameter $\beta$ defines the score-prioritized distribution $P_S(\Lambda_{\text{train}})$ over the training levels, under which

$$P_S(l_i|\Lambda_{\text{seen}}, S) = \frac{h(S_i)^{1/\beta}}{\sum_j h(S_j)^{1/\beta}}. \tag{3}$$

The function $h$ defines how differences in level scores translate into differences in prioritization. The temperature parameter $\beta$ allows us to tune how much $h(S)$ ultimately determines the resulting distribution. We make the design choice of using rank prioritization, for which $h(S_i) = 1/\text{rank}(S_i)$, where $\text{rank}(S_i)$ is the rank of level score $S_i$ among all scores sorted in descending order. We also experimented with proportional prioritization ($h(S_i) = S_i$), which performs comparably to rank prioritization, as well as greedy prioritization (the level with the highest score receives probability 1), which tends to perform worse.

## 3.2 STALENESS-AWARE PRIORITIZATION

As the scores used to parameterize $P_S$ are a function of the state of the policy at the time the associated level was last played, they come to reflect a gradually more off-policy measure the longer they remain without an update through replay. We mitigate this drift towards "off-policy-ness" by explicitly mixing the sampling distribution with a staleness-prioritized distribution $P_C$

$$P_C(l_i|\Lambda_{\text{seen}}, C, c) = \frac{c - C_i}{\sum_{C_j \in C} c - C_j} \tag{4}$$

which assigns probability mass to each level $l_i$ in proportion to the level's *staleness* $c - C_i$. Here, $c$ is the count of total episodes sampled so far in training and $C_i$ (referred to as the level's timestamp) is the episode count at which $l_i$ was last sampled. By pushing support to levels with staler scores, $P_C$ ensures no score drifts too far off-policy.

Plugging Equations 3 and 4 into Equation 1 gives us a replay distribution that is calculated as

$$P_{\text{replay}}(l_i) = (1 - \rho) \cdot P_S(l_i|\Lambda_{\text{seen}}, S) + \rho \cdot P_C(l_i|\Lambda_{\text{seen}}, C, c).$$

Thus, a level has an increased chance of being sampled when its score is high or it has not been sampled for a long time.

## 4 EXPERIMENTAL SETTING

We evaluate the effectiveness of Prioritized Level Replay on several PCG environments with various implementations of scoring functions and prioritization schemes, and compare against the most common direct level sampling baseline for which $P_{\text{train}}(l|\Lambda_{\text{train}}) = \textbf{Uniform}(l; \Lambda_{\text{train}})$. Specifically, we train and test on all 16 games in the Procgen Benchmark on the easy difficulty setting (Cobbe et al., 2019) and select hard exploration environments from the MiniGrid suite (Chevalier-Boisvert et al., 2018). In addition, we demonstrate that our method, when combined with UCB-DrAC (Raileanu et al., 2020), sets a new state-of-the-art on Procgen Benchmark. In line with the standard baseline for these environments, all our experiments use PPO with GAE for training. For Procgen Benchmark we use the same ResBlock architecture as Cobbe et al. (2019) and the 200 training levels used in the original baselines, while sampling unseen levels for every test run. For MiniGrid, we use a 3-layer CNN architecture similar to that in Igl et al. (2019), and provide approximately 1000 levels of each difficulty for every environment during training. Detailed descriptions of the environments, architectures, and hyperparameters used in our experiments (as well as how they were set or obtained) can be found in Appendices A and B. See Table 1 for the full set of scoring functions $\textbf{score}(\tau, \pi)$ investigated in our experiments. Additionally, we report results of training on the full level distribution of select MiniGrid levels in Appendix C.2.

Table 1: Choices of scoring function $\mathbf{score}(\tau, \pi)$ investigated in this work.

| Scoring metric | $\mathbf{score}(\tau, \pi)$ |
|---|---|
| Policy entropy | $\frac{1}{T} \sum_{t=0}^{T} \sum_a \pi(a, s_t) \log \pi(a, s_t)$ |
| Policy min-margin | $\frac{1}{T} \sum_{t=0}^{T} (\max_a \pi(a, s_t) - \max_{a \neq \max_a \pi(a, s_t)} \pi(a, s_t))$ |
| Policy least-confidence | $\frac{1}{T} \sum_{t=0}^{T} (1 - \max_a \pi(a, s_t))$ |
| Unsigned 1-step TD error | $\frac{1}{T} \sum_{t=0}^{T} |\delta_t|$ |
| GAE | $\frac{1}{T} \sum_{t=0}^{T} \sum_{k=t}^{T} (\gamma\lambda)^{t-k} \delta_t$ |
| Unsigned GAE (L1 value loss) | $\frac{1}{T} \sum_{t=0}^{T} \left| \sum_{k=t}^{T} (\gamma\lambda)^{t-k} \delta_t \right|$ |

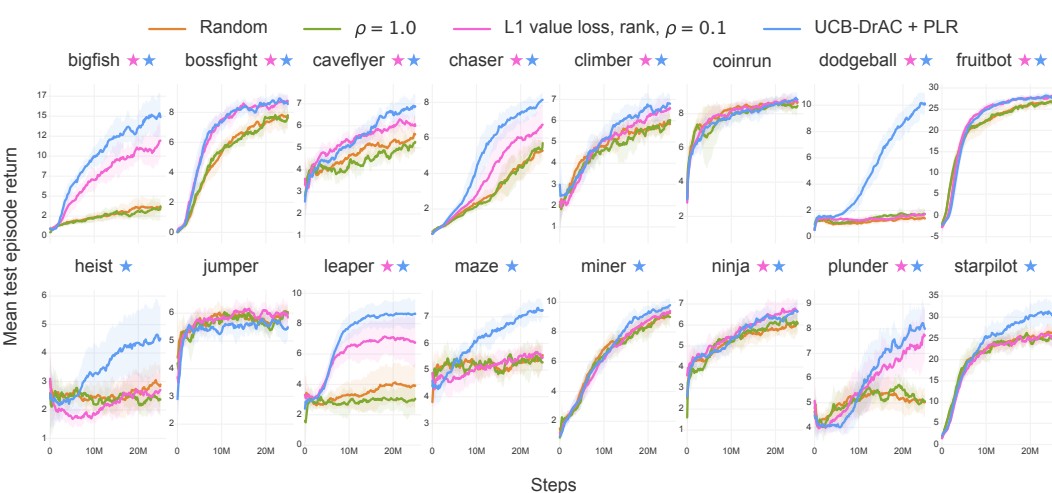

Figure 2: Mean episodic test returns (10 runs) while training under value-based level replay with rank prioritization, both with and without UCB-AutoDrAC, compared to baseline sampling methods. Each colored ★ indicates statistically significant ($p = 0.05$) gains in final test performance or sample complexity along the curve, relative to uniform sampling for the condition of the same color.

## 5 RESULTS AND DISCUSSION

Prioritized Level Replay consistently provides better (or at least on-par) results than direct level sampling. In particular, we find that: (i) value-based level replay consistently outperforms direct level sampling in generalization and sample-efficiency across environments, (ii) alternative scoring functions lead to inconsistent improvements across environments, (iii) our method can yield significant additive gains combined with other methods, and (iv) value-based level replay induces an implicit curriculum over training levels, which substantially aids training in hard exploration environments.

### 5.1 PROCGEN BENCHMARK

Our results in Figure 2 show that value-based level replay combined with rank prioritization, with $\beta = 0.1$ and $\rho = 0.1$, leads to the most drastic improvements across games. This setting yields statistically significant ($p = 0.05$) improvements in generalization performance over uniform-sampling on 10 out of 16 environments, according to Welch's t-test. In comparison, value-based level replay with proportional prioritization provides statistically significant gains over uniform sampling on one more game, but on average sees worse mean test returns. Notably, gains only occur when the replay distribution considers both level scores and staleness, highlighting the importance of staleness-based sampling in keeping scores from drifting off-policy. Further, we report considerable improvements over uniform level-sampling on the hard setting, when directly using the best hyperparameters found for the easy setting. Figures 8, 9, 11 and Table 4 in Appendix C report these additional results.

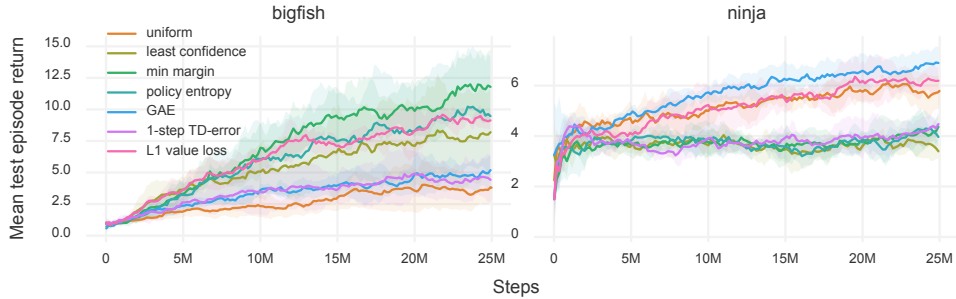

Figure 3: Two example Procgen games, between which all scoring functions except for L1 value loss show inconsistent improvements to generalization and sample efficiency, with rank prioritization $\beta = 0.1$, and $\rho = 0.3$. This inconsistency held across settings in our grid search.

The remaining scoring metrics based on TD-error and classifier uncertainty perform inconsistently across games. While certain games such as BigFish seem amenable to improvements in both training sample complexity and generalization under various Prioritized Level Replay scoring metrics, other games, such as Ninja, see no improvement or even degraded sample complexity and generalization. See Figure 3 for an example of this inconsistent effect across games.

We conjecture that sampling levels with Prioritized Level Replay under value-based replay induces an implicit curriculum over the training levels. Easier levels result in non-zero, non-stationary returns earlier in training, while harder levels give rise to near stationary returns until the agent learns an improved policy that allows making further progress on the level. Therefore, meaningful non-stationary value regression targets appear earlier in easier levels than in harder levels, leading to higher value losses on easier levels than on harder levels, where the agent can not yet make progress. Sampling levels according to the L1 value-loss then leads to an implicit curriculum from easier to harder levels. In the next section, we test this conjecture in hard exploration environments in Mini-Grid, whose levels are clearly separated into discrete, qualitatively different difficulties.

Prioritized Level Replay easily combines with most other methods, as it simply defines a process for adapting the training level-sampling distribution as a function of the agent's past trajectories and policy instantiations during those trajectories. To demonstrate this fact, we combine our method with UCB-DrAC (Raileanu et al., 2020). At the time of writing, UCB-DrAC had attained state-of-the-art results on the Procgen Benchmark. Our combined setup trains agents using UCB-DrAC while sampling training levels at the start of each episode according to value-based level replay. As summarized in Table 2, UCB-DrAC combined with our method sets a new state-of-the-art on the Procgen Benchmark. Figure 2 showcases the test performance throughout training of this combined approach. The full details of this experiment are provided in Appendix C.3.

## 5.2 MINIGRID

As in the case of the Procgen Benchmark, we find that value-based level replay with rank prioritization significantly improves the sample efficiency and generalization performance in both environments. On these MiniGrid environments, we find a slightly higher staleness coefficient of $\rho = 0.3$ leads to better test performance. The results are summarized in the top row of Figure 4.

For evaluation purposes, we bin each level into its corresponding difficulty, expressed as ascending, discrete values (note that Prioritized Level Replay does not have access to this privileged information). From Figure 4 we see how the expected difficulty of levels sampled using value-based level replay changes throughout training for each environment. We observe that as level weights are updated based on the agent's past experiences of each level, these levels become sampled according to an implicit curriculum over the training levels that prioritizes progressively harder levels. Notably, value-based level replay struggles to discover a useful curriculum for around the first 4,000 updates on ObstructedMazeGamut-Medium, at which point it discovers a curriculum that gradually assigns more weight to harder levels. This curriculum appears quite effective, enabling value-based level replay with access to only 6,000 training levels to attain even higher mean test returns than the uniform-sampling baseline with access to the full set of training levels, of which there are roughly 4 billion by our estimate (so our training levels constitute 0.00015% of the total number).

Table 2: Comparison of test performance of policies trained on each method. Here PLR denotes value-based level replay with rank prioritization with $\beta = 0.1$ and $\rho = 0.1$. Following the evaluation protocol in Raileanu et al. (2020), the reported mean and standard deviation per environment is computed by evaluating the final policy's average return on 100 test episodes, aggregated across multiple training runs (10 runs for Procgen Benchmark and 3 for MiniGrid). Each run is initialized with a different training seed. Normalized test returns per run are computed by dividing the average test return per run for each environment by the corresponding average test return of the uniform-sampling baseline over all runs. We then report the means and standard deviations of normalized test returns aggregated across runs. Note, we report the normalized return statistics for Procgen and MiniGrid environments separately.

| Environment | Uniform | UCB-DrAC | PLR | UCB-DrAC+PLR |
|---|---|---|---|---|
| BigFish | $3.72 \pm 1.24$ | $8.73 \pm 1.13$ | $10.91 \pm 2.81$ | $\mathbf{14.28 \pm 2.12}$ |
| BossFight | $7.7 \pm 0.37$ | $7.65 \pm 0.67$ | $\mathbf{8.94 \pm 0.35}$ | $8.84 \pm 0.8$ |
| CaveFlyer | $5.37 \pm 0.79$ | $4.61 \pm 0.93$ | $6.32 \pm 0.47$ | $\mathbf{6.76 \pm 0.7}$ |
| Chaser | $5.23 \pm 0.69$ | $6.79 \pm 0.93$ | $6.9 \pm 1.21$ | $\mathbf{8.01 \pm 0.6}$ |
| Climber | $5.93 \pm 0.6$ | $6.39 \pm 0.92$ | $6.33 \pm 0.84$ | $\mathbf{6.8 \pm 0.67}$ |
| CoinRun | $8.62 \pm 0.4$ | $8.62 \pm 0.45$ | $8.76 \pm 0.47$ | $\mathbf{8.95 \pm 0.37}$ |
| Dodgeball | $1.69 \pm 0.23$ | $5.11 \pm 1.65$ | $1.78 \pm 0.46$ | $\mathbf{10.33 \pm 1.36}$ |
| FruitBot | $27.29 \pm 0.94$ | $27.02 \pm 1.35$ | $\mathbf{28.02 \pm 1.35}$ | $27.62 \pm 1.47$ |
| Heist | $2.77 \pm 0.92$ | $3.17 \pm 0.74$ | $2.93 \pm 0.48$ | $\mathbf{4.93 \pm 1.3}$ |
| Jumper | $5.71 \pm 0.42$ | $5.61 \pm 0.46$ | $5.83 \pm 0.48$ | $\mathbf{5.86 \pm 0.34}$ |
| Leaper | $4.18 \pm 1.33$ | $4.44 \pm 1.42$ | $6.83 \pm 1.15$ | $\mathbf{8.66 \pm 0.98}$ |
| Maze | $5.46 \pm 0.37$ | $6.21 \pm 0.5$ | $5.49 \pm 0.8$ | $\mathbf{7.23 \pm 0.82}$ |
| Miner | $8.73 \pm 0.72$ | $\mathbf{10.09 \pm 0.6}$ | $9.56 \pm 0.62$ | $10.03 \pm 0.54$ |
| Ninja | $6.04 \pm 0.41$ | $5.83 \pm 0.79$ | $\mathbf{7.24 \pm 0.38}$ | $6.96 \pm 0.5$ |
| Plunder | $5.05 \pm 0.55$ | $7.79 \pm 0.89$ | $\mathbf{8.68 \pm 2.18}$ | $7.67 \pm 0.95$ |
| StarPilot | $26.84 \pm 1.54$ | $\mathbf{31.68 \pm 2.36}$ | $27.9 \pm 4.35$ | $29.64 \pm 2.22$ |
| Normalized Returns (%) | $100.0 \pm 4.5$ | $129.8 \pm 8.2$ | $128.3 \pm 5.8$ | $\mathbf{176.4 \pm 6.1}$ |
| MultiRoom-N4-Random | $0.80 \pm 0.04$ | – | $\mathbf{0.81 \pm 0.01}$ | – |
| ObstructedMazeGamut-Easy | $0.53 \pm 0.04$ | – | $\mathbf{0.85 \pm 0.04}$ | – |
| ObstructedMazeGamut-Medium | $0.65 \pm 0.01$ | – | $\mathbf{0.73 \pm 0.07}$ | – |
| Normalized Returns (%) | $100.0 \pm 2.5$ | – | $\mathbf{124.3 \pm 4.7}$ | – |

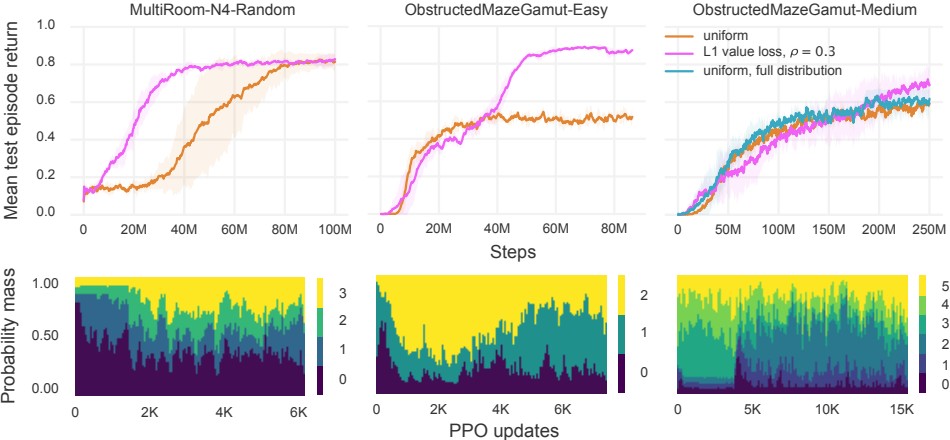

Figure 4: Top: Mean episodic test returns of value-based level replay and the uniform-sampling baseline on MultiRoom-N4-Random (4 runs), ObstructedMazeGamut-Easy (3 runs), and ObstructedMazeGamut-Medium (3 runs). Bottom: The probability mass assigned to levels of varying difficulty over the course of training for the respective environment for a single run.

## 6 RELATED WORK

Prior work for improving generalization in deep RL adapt techniques from supervised learning, including stochastic regularization (Igl et al., 2019; Cobbe et al., 2019), data augmentation (Kostrikov et al., 2020; Raileanu et al., 2020), and knowledge distillation (Igl et al., 2020; Cobbe et al., 2020). In contrast, Prioritized Level Replay requires no modifications to the model architecture or training algorithm—it only modifies the order in which rollouts are sampled from a collection of training levels. Our method is compatible with any model or policy-gradient algorithm, and can be implemented alongside these approaches for potentially orthogonal gains.

Prioritized Level Replay is related to active learning (Cohn et al., 1994; Settles, 2009) and echoes ideas from Graves et al. (2017), which uses gradient-based progress signals for training an adversarial multi-armed bandit that selectively samples the next task in multi-task supervised learning. Sharma et al. (2018) extend these ideas to multi-task RL, but add the additional requirement of knowing a maximum target return for each task a priori. More recently, Zhang et al. (2020) use an ensemble of value functions for selective goal sampling in the off-policy continuous control setting, requiring prior knowledge of the environment structure to generate candidate goals. Notably, these methods depend on the ability to sample instances of specific similarly-structured tasks, an assumption that breaks in the PCG setting. Rather, we show that our method automatically uncovers similarly difficult levels, inducing a curriculum without any prior knowledge of the environment.

A recent theme in automated curriculum learning explores adaptively modifying the environment to facilitate learning. Khalifa et al. (2020) and Wang et al. (2019; 2020) introduce methods that jointly generate levels and train agents to solve them, leading to a co-evolving curriculum. Like Prioritized Level Replay, such curricula exploit the changing differences in learning potential across levels throughout training. Unlike these approaches, our method does not assume control over level generation, requiring only the ability to replay previously visited levels, as referenced by a level identifier such as the corresponding seed used by the PCG algorithm to generate the level. A related set of approaches makes use of multi-agent RL to train a separate teacher agent, which learns a goal or task proposal policy (Matiisen et al., 2020; Campero et al., 2020; Sukhbaatar et al., 2017). In contrast to such teacher-student auto-curriculum methods, our approach relies only on a single agent, and therefore does not require extending the environment to accommodate additional agents, such as including teacher-specific action spaces.

Like our method, Schaul et al. (2015) uses TD-errors as a signal for learning potential. However, while their method only scores 1-step TD-errors to assess the value of revisiting individual past transitions, our method uses estimators based on $k$-step TD errors to predict the relative value of different choices for generating and training on entire trajectories of future experiences.

## 7 CONCLUSION AND FUTURE WORK

We introduced Prioritized Level Replay, a method for improving sample-efficiency and generalization performance of RL agents trained and evaluated on procedurally generated environments. Prioritized Level Replay updates level sampling weights according to a scoring metric throughout the course of training. We surveyed the effectiveness of several variations of our method to find that L1 value-loss scores consistently lead to statistically significant improvements in sample efficiency and generalization on a wide variety of PCG games, including the majority of environments in the ProcGen Benchmark and two challenging MiniGrid environments. Our method combines with a prior leading method to improve the state-of-the-art on Procgen Benchmark. Further, on MiniGrid environments, we observe that our method induces an implicit curriculum over the training levels.

While we evaluated Prioritized Level Replay on PCG environments, we believe our method may be generally applicable to any RL environment, given a procedure for generating novel variations of the underlying MDP as a function of a level identifier, for example by varying the starting positions of game entities. Furthermore, as our method modifies only how levels are sampled per episode, it combines easily with any on-policy RL algorithm, as well as other methods for improving sample complexity and generalization, potentially leading to additive gains. Lastly, the scoring metric can be more generally parameterized as a learned function of the level and statistics of past trajectories over that level, which may prove even more effective than the L1 value-loss. We plan to investigate these promising directions in future work.

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

# A    EXPERIMENT DETAILS

## A.1    PROCGEN BENCHMARK

The OpenAI ProcGen Benchmark suite consists of 16 procedurally-generated games of varying styles, exhibiting a diversity of gameplay similar to that of the ALE benchmark. Levels in each game are determined by a seed, and can vary in many ways, including layout, visual appearance, and starting positions of game entities. All ProcGen environments share the same discrete 15-dimensional action space and produce $64 \times 64 \times 3$ RGB observations. Cobbe et al. (2019) provides a comprehensive description of each of the 16 environments.

We follow the original protocol for testing generalization performance on ProcGen defined by Cobbe et al. (2019): We train an agent for each game on a finite number of levels, $N_{\text{train}}$, and test on the full distribution of levels. Previous benchmarks showed that training under this protocol using state-of-the-art policy-gradient algorithms leads to a significant generalization gap between test and train performance in all games, making ProcGen a useful benchmark for assessing generalization performance.

To make the most efficient use of our computational resources, we train on the easy difficulty mode using the recommended settings of $N_{\text{train}} = 200$ and 25M steps of training, as well as the same ResNet policy architecture and PPO hyperparameters shared across all games as Cobbe et al. (2019). We find 25M steps to be sufficient for uncovering differences in generalization performance among our methods and standard baselines. Moreover, under this setup, we find ProcGen training runs require much less wall-clock time than training runs on our MiniGrid environments of interest over an equivalent number of steps needed to uncover differences in generalization performance. Therefore we perform a survey of the empirical differences of various settings of Prioritized Level Replay on ProcGen rather than MiniGrid.

We evaluate each combination of the scoring function choices in Table 1 with both rank and proportional prioritization, performing a coarse grid search for each pair over different settings of the temperature parameter $\beta$ in $\{0.1, 0.5, 1.0, 1.4, 2.0\}$ and the staleness coefficient $\rho$ in $\{0.1, 0.3, 1.0\}$. For each setting, we run 4 trials across all 16 of games of the Procgen Benchmark. We find that rank prioritization tends to produce larger improvements to test performance than proportional prioritization, and the best setting across Procgen games was $\beta = 0.1$, $\rho = 0.1$. In our TD-error-based scoring functions, we set $\gamma$ and $\lambda$ equal to the same respective values used by the GAE in PPO during training. See Appendix B for an comprehensive overview of the same hyperparameter setting we used for PPO across all Procgen environments.

## A.2    MINIGRID

In order to validate the conjecture that Prioritized Level Replay with L1 value-loss scores induces an implicit curriculum over the training levels, we turn to the MiniGrid suite (Chevalier-Boisvert et al., 2018). MiniGrid features a series of highly structured environments of increasing difficulty. Each environment features a task in a gridworld setting, and as in ProcGen, environment levels are determined by a seed. Harder levels require the agent to perform longer action sequences over a combinatorially-rich set of game entities, on increasingly larger grids. The clear ordering of difficulty over subsets of MiniGrid environments allows us to track the relative difficulty of levels sampled under Prioritized Level Replay over the course of training.

MiniGrid environments share a discrete 7-dimensional action space and produce a 3-channel integer state encoding of the $7 \times 7$ grid immediately including and in front of the agent. However, following the training setup in Igl et al. (2019), we modify the environment to produce an $N \times M \times 3$ encoding of the full grid, where $N$ and $M$ vary according to the maximum grid dimensions of each environment. This fully observable setup requires the agent to generalize across different level layouts.

We evaluate value-based level replay with rank prioritization on two distinct MiniGrid environments, whose levels are uniformly distributed across several difficulty settings. Training on levels of varying difficulties helps agents make use of the easier levels as stepping stones to learn useful behaviors that help the agent make progress on harder levels. However, under the uniform-sampling baseline, learning may be inefficient, as the training process does not selectively train the agent on levels of increasing difficulty, leading to wasted training steps when a difficult level is sampled

early in training. On the contrary, if value-based level replay scores samples levels according to the time-averaged L1 value loss of recently experienced level trajectories, the average difficulty of the sampled levels should increase over time, following the reasoning outlined in 5.1, that significant non-stationary value targets will be encountered earlier in easier levels.

As in Igl et al. (2019), we parameterize the agent policy as a 3-layer CNN with 16, 32, and 32 channels. with a final hidden layer of size 64. All kernels are $2 \times 2$ and use a stride of 1. For the ObstructedMazeGamut environments, we increase the number of channels of the final CNN layer to 64. We follow the same generalization evaluation protocol as used for ProcGen, training the agent on a fixed set of 4000 levels for MultiRoom-N4-Random, 3000 levels for ObstructedMazeGamut-Easy, and 6000 levels for ObstructedMazeGamut-Medium, and testing on the full level distribution. We chose these values for $\Lambda_{\text{train}}$ to ensure roughly 1000 training levels of each represented difficulty setting of each environment. As in the case of our Procgen experiments, we we set $\gamma$ and $\lambda$ equal to the same respective values used by the GAE in PPO during training, and use reward normalization during training. All MiniGrid experiments shared the same hyperparameters listed in B and, when using value-based level replay, set $\beta = 0.1$, $\rho = 0.3$ with rank prioritization.

The remainder of this section provides more details about the various MiniGrid environments used in this work.

**MultiRoom-N4-Random**   This environment requires the agent to navigate through 1, 2, 3, or 4 rooms respectively to reach a goal object, resulting in a natural ordering of levels over four levels of difficulty. The agent always starts at a random position in the furthest room from the goal object, facing a random direction. The goal object is also initialized to a random position within its room. See Figure 5 for screenshots of example levels.

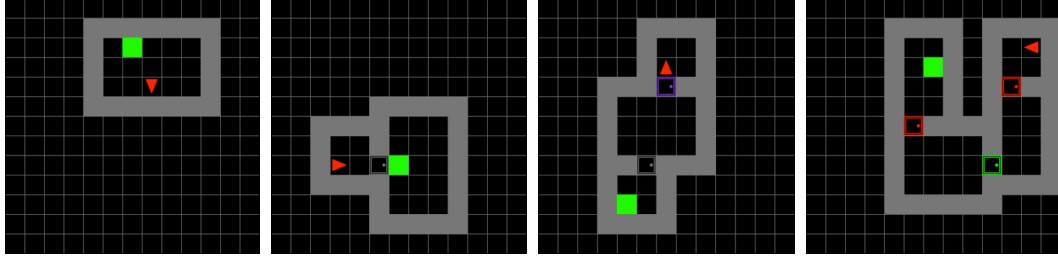

Figure 5: Example levels of each of the four difficulty levels of MultiRoom-N4-Random, in order of increasing difficulty from left to right. The agent (red triangle) must reach the goal (green square).

**ObstructedMaze-Gamut-Easy**   This environment consists of levels uniformly distributed across the first three difficulty settings of the ObstructedMaze environment, in which the agent must locate and pick-up the key in order to unlock the door to pick-up a goal object in a second room. The agent and goal object are always initialized in random positions in different rooms separated by the locked door. The second difficulty setting further requires the agent to first uncover the key from under a box before picking up the key. The third difficulty level further requires the agent to first move a ball blocking the door before entering the door. See Figure 6 for screenshots of example levels.

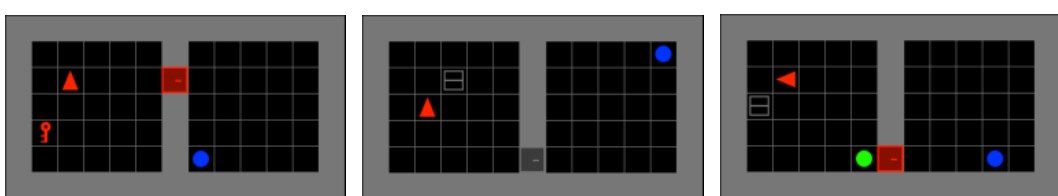

Figure 6: Example levels of each of the three difficulty levels of ObstructedMaze-Gamut-Easy, in order of increasing difficulty from left to right. The agent must find the key, which may be hidden under a box, to unlock a door granting access to the goal object (blue circle).

**ObstructedMaze-Gamut-Hard** This environment consists of levels uniformly distributed across the first six difficulty levels of the ObstructedMaze environment. Harder levels corresponding to the fourth, fifth, and sixth difficulty settings include two additional rooms with no goal object to distract the agent. Each instance of these harder levels also contain two pairs of keys of different colors, each opening a door of the same color. The agent always starts one room away from the randomly positioned goal object. Each of the two keys is visible in the fourth difficulty setting and doors are unobstructed. The fifth difficulty setting hides the keys under boxes, and the sixth again places obstacles that must be removed before entering two of the doors, one of which is always the door to the goal-containing room. See Figure 7 for example screenshots.

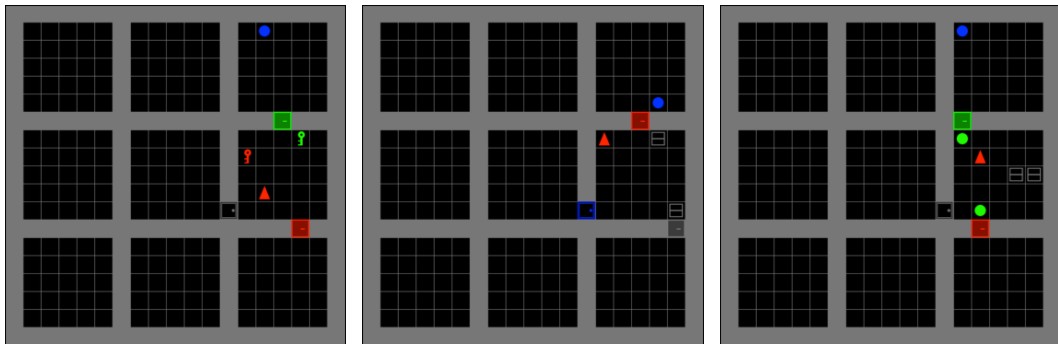

Figure 7: Example levels in increasing difficulty from left to right of each additional difficulty setting introduced by ObstructedMaze-Gamut-Hard in addition to those in ObstructedMaze-Gamut-Easy.

## B  HYPERPARAMETERS

Table 3 lists the hyperparameters used during training. The values in each column were shared across all environments in Procgen and MiniGrid respectively. For Procgen, these are the defaults used by Cobbe et al. (2019). We adapted these for Minigrid, but did no further grid-searching of these values.

Table 3: Hyperparameters used for training.

| Parameter | Procgen | MiniGrid |
|---|---|---|
| $\gamma$ | 0.999 | 0.999 |
| $\lambda_{\text{GAE}}$ | 0.95 | 0.95 |
| PPO rollout length | 256 | 256 |
| PPO epochs | 3 | 4 |
| PPO minibatches per epoch | 8 | 8 |
| PPO clip range | 0.2 | 0.2 |
| PPO number of workers | 64 | 64 |
| Adam learning rate | 5e-4 | 7e-4 |
| Adam $\epsilon$ | 1e-5 | 1e-5 |
| reward normalization | yes | yes |
| entropy bonus coefficient | 0.01 | 0.01 |
| value loss coefficient | 0.5 | 0.5 |

## C  ADDITIONAL EXPERIMENTAL RESULTS

### C.1  EXTENDED RESULTS ON PROCGEN BENCHMARK

We show in Figures 8 and 9, the mean test episodic returns on the Procgen Benchmark for value-based level replay with rank and proportional prioritization, respectively. In both of these plots, we

can see that using only staleness ($\rho = 1$) or only L1 value loss scores ($\rho = 0$) is considerably worse than direct level sampling. Thus, we only observe gains compared to the baseline when both level scores and staleness are used for the sampling distribution. Comparing Figures 8 with 9 we find that value-based level replay with proportional instead of rank prioritization provides more statistically significant gains compared to direct level sampling, but on average leads to worse mean test returns than rank prioritization.

Lastly, we show in Figure 10 that when value-based level replay improves generalization performance on a game, it also either matches or improves training sample efficiency. This shows that when beneficial to test-time performance, the representations learned via the auto-curriculum induced by our method prove similarly useful on the training levels. However we see that our method reduces training sample efficiency on two games on which our method does not improve generalization performance. Since our method does not discover useful auto-curricula for these games, it is likely that uniformly sampling levels at training time allows the agent to better memorize useful behaviors on each of the training levels compared to the selective sampling performed by our method.

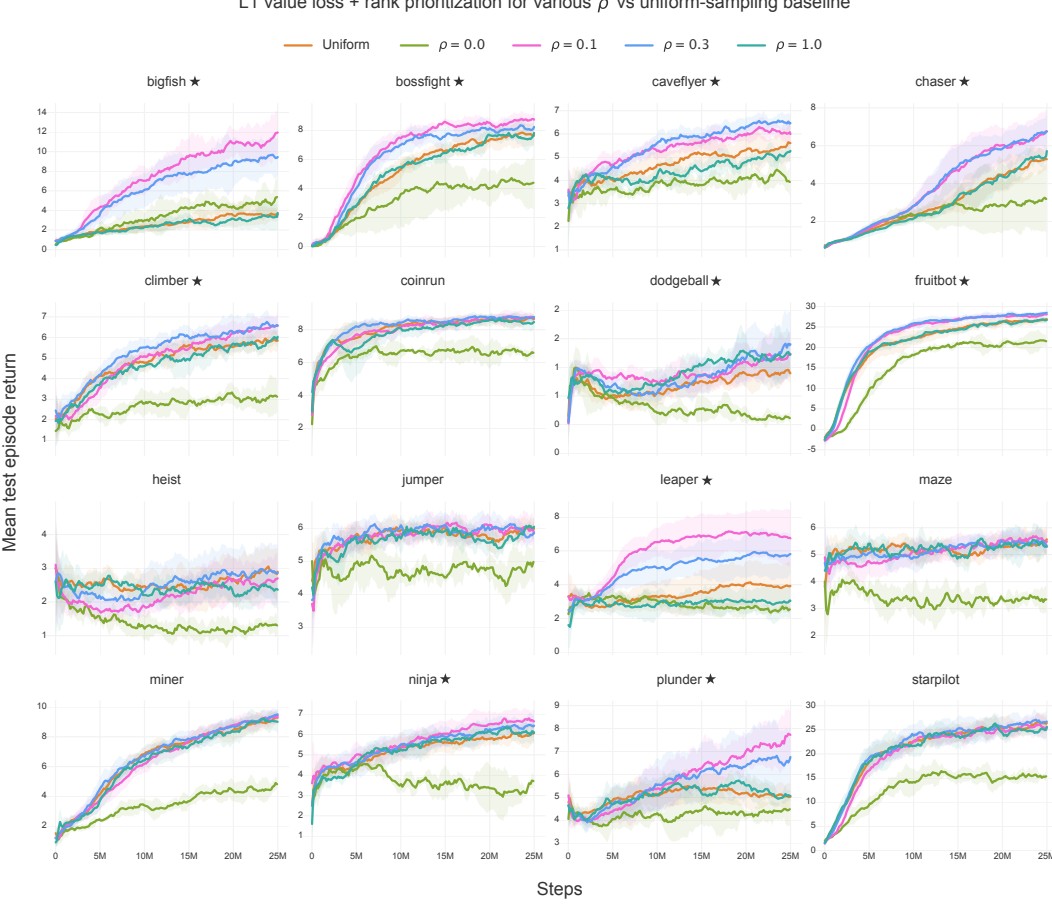

Figure 8: Mean test episodic returns on the Procgen Benchmark for value-based level replay with rank prioritization with $\beta = 0.1$ across a range of values of $\rho$, which weighs how much importance the final replay distribution assigns to staleness. The replay distribution must consider both the L1 value-loss and staleness values to realize improvements to generalization and sample complexity. Plots are averaged over 10 runs, and the shaded area indicates one standard deviation around the mean. A ★ next to the game name indicates the condition $\rho = 0.1$ exhibits statistically significantly better final test returns or sample efficiency along the test curve at $p = 0.05$, , which we observe in 10 of 16 games.

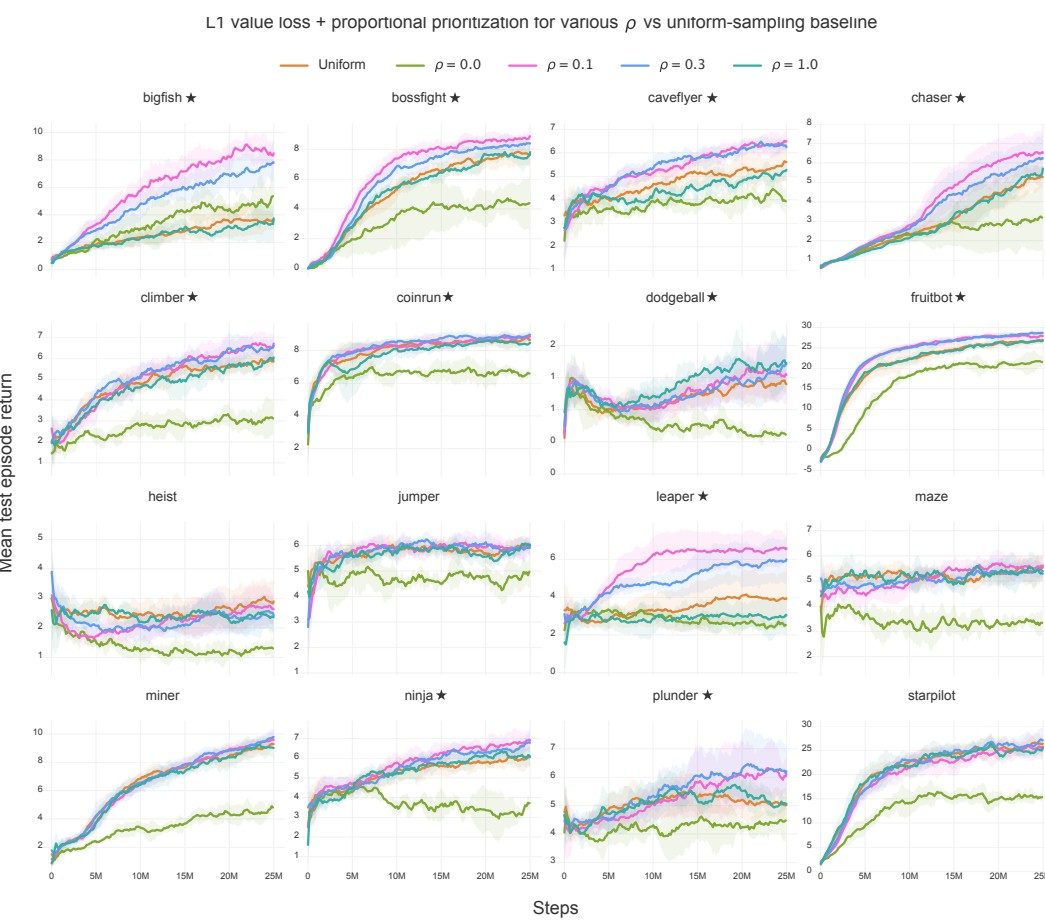

Figure 9: Mean test episodic returns on the Procgen Benchmark for value-based level replay with proportional prioritization with $\beta = 0.1$ across a range of values of $\rho$. As in the case of rank prioritization, the replay distribution must consider both the L1 value loss score and level staleness values in order to realize performance improvements. Plots are averaged over 10 runs, and the shaded area indicates one standard deviation around the mean. A ★ next to the game name indicates the condition $\rho = 0.1$ exhibits statistically significantly better final test returns or sample efficiency along the test curve at $p = 0.05$, which we observe in 11 of 16 games.

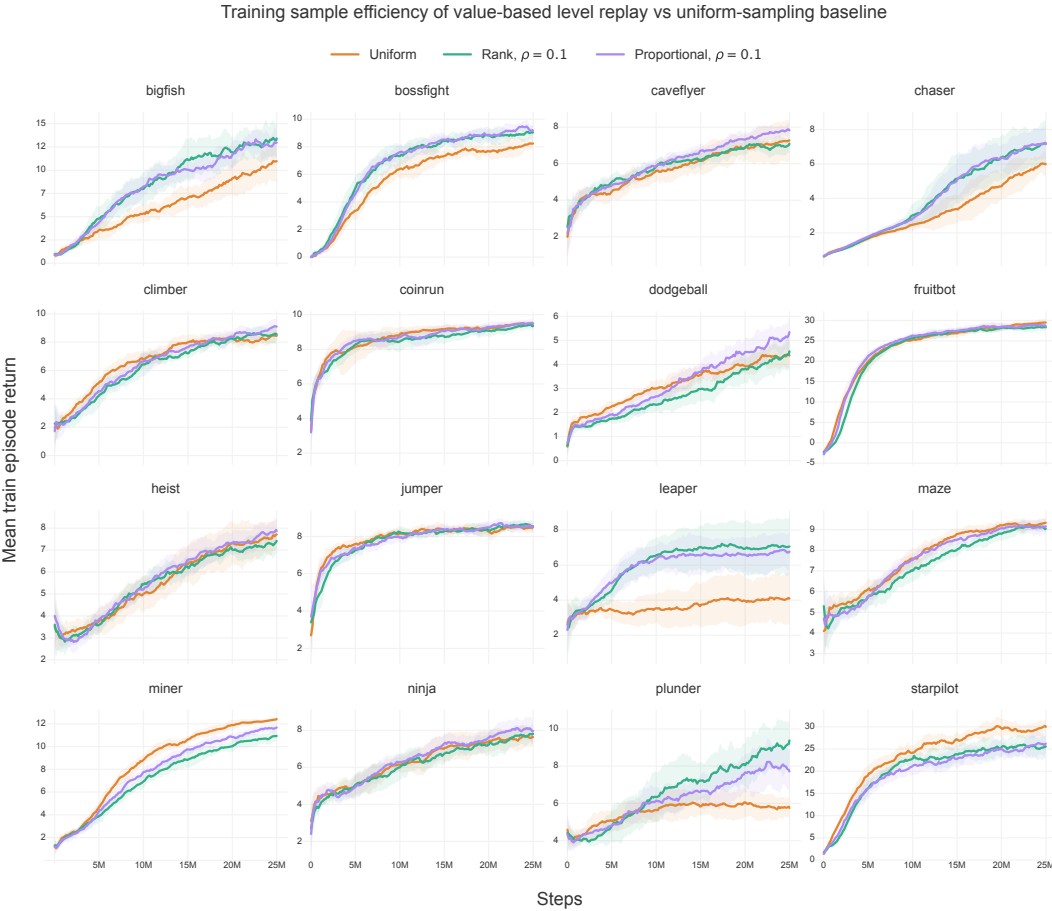

Figure 10: Mean train episodic returns on the Procgen Benchmark for value-based level replay with $\beta = 0.1$ and each of the prioritization methods investigated. On some games, value-based replay improves both training sample efficiency and generalization performance (e.g. BigFish and Chaser), while on others, only generalization performance (e.g. CaveFlyer for rank prioritization). Plots are averaged over 10 runs, and the shaded area indicates one standard deviation around the mean.

Table 4: Comparison of test scores of PPO with value-based level replay (PLR) against PPO with uniform-sampling on the hard setting of Procgen Benchmark. As in Table 2, reported figures represent the mean and standard deviation of average test scores over 100 episodes aggregated across 10 runs, each initialized with a unique training seed. For each run, a normalized average return is computed by dividing the average test return for each game by the corresponding average test return of the uniform-sampling baseline over all 1000 test episodes of that game, followed by averaging these normalized returns over all 16 games. The final row reports the mean and standard deviation of the normalized returns aggregated across runs. Note that while the uniform-sampling baseline achieves a higher mean return on BigFish and StarPilot than value-based level replay, this difference was not found to be statistically significant according to Welch's t-test.

| Games | Uniform | PLR |
|---|---|---|
| BigFish | $\mathbf{9.13 \pm 4.51}$ | $7.77 \pm 1.01$ |
| BossFight | $6.82 \pm 0.59$ | $\mathbf{8.67 \pm 0.72}$ |
| CaveFlyer | $3.13 \pm 0.47$ | $\mathbf{6.36 \pm 0.08}$ |
| Chaser | $5.3 \pm 1.22$ | $\mathbf{6.26 \pm 0.67}$ |
| Climber | $3.26 \pm 0.46$ | $\mathbf{6.23 \pm 0.76}$ |
| CoinRun | $5.06 \pm 0.24$ | $\mathbf{5.42 \pm 0.39}$ |
| Dodgeball | $1.76 \pm 0.29$ | $\mathbf{2.01 \pm 1.09}$ |
| FruitBot | $11.15 \pm 2.59$ | $\mathbf{15.86 \pm 1.26}$ |
| Heist | $0.84 \pm 0.36$ | $\mathbf{1.24 \pm 0.37}$ |
| Jumper | $3.3 \pm 0.46$ | $\mathbf{3.58 \pm 0.46}$ |
| Leaper | $2.52 \pm 1.45$ | $\mathbf{6.42 \pm 0.43}$ |
| Maze | $3.98 \pm 0.18$ | $\mathbf{4.12 \pm 0.46}$ |
| Miner | $9.5 \pm 0.15$ | $\mathbf{9.67 \pm 0.44}$ |
| Ninja | $3.14 \pm 0.33$ | $\mathbf{5.36 \pm 0.52}$ |
| Plunder | $2.72 \pm 0.34$ | $\mathbf{4.1 \pm 1.32}$ |
| StarPilot | $\mathbf{2.85 \pm 0.7}$ | $2.63 \pm 0.3$ |
| Normalized Return (%) | $100.0 \pm 9.5$ | $\mathbf{138.6 \pm 9.6}$ |

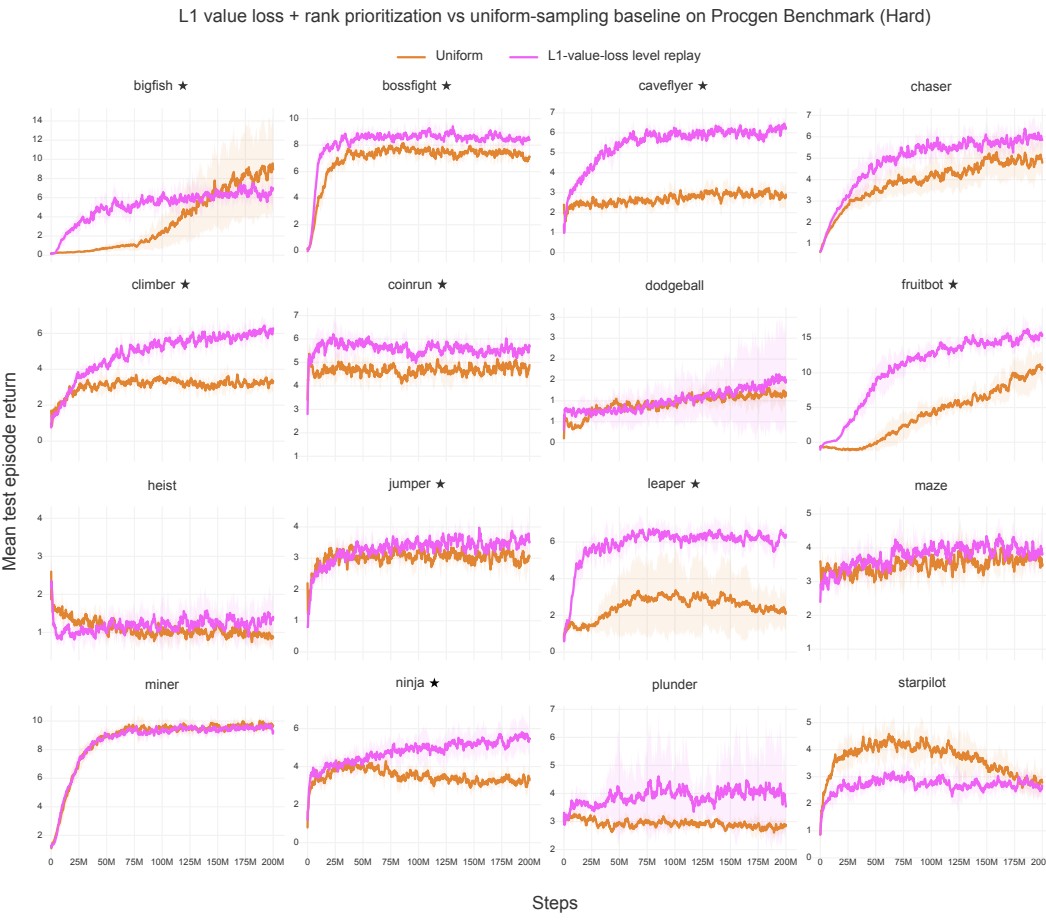

Figure 11: Comparison of mean train episodic returns on Procgen Benchmark's hard setting for value-based level replay against the uniform-sampling baseline. These results were generated with the best hyperparameters for value-based level replay found for Procgen Benchmark's easy setting, where $\beta = \rho = 0.01$. Plots are averaged over 5 runs, and the shaded area indicates one standard deviation around the mean. Note that even without tuning level-replay hyperparameters for Procgen Benchmark's hard setting, we reach statistically significant improvements in test performance on the majority of the benchmark (9 of 16 games).

## C.2 TRAINING ON THE FULL LEVEL DISTRIBUTION

While assessment of generalization performance calls for using a fixed set of training levels, ideally our method can also make use of the full level distribution if given access to it. To take advantage of an unspecified number of training levels sampled from the full level distribution, we modify the list structures for storing scores and timestamps (see Algorithm 1 and 2) to keep up to a maximum of level buffer size, $M$, items. When the lists are full, we set the next level for replacement to be

$$l_{\min} = \arg\min_l P_{\text{replay}}(l).$$

When the outcome of the Bernoulli $P_D$ entails sampling from the replay distribution, resulting in sampling a new level $l$, the score and timestamps of $l$ replace those of $l_{\min}$ only if the score of $l_{\min}$ is lower than that of $l$. In this way, Prioritized Level Replay keeps a running buffer throughout training of the top $M$ levels appraised to have the highest learning potential for replaying anew.

Figure 12 shows that with access to the full level distribution at training, value-based level replay improves sample-efficiency and generalization performance in both environments compared to uniform sampling. In MultiRoom-N4-Random, the value $M$ makes little difference to generalization performance, and access to the full level distribution at training leads to a policy outperforming one trained with value-based level replay on a fixed set of training levels. However, on ObstructedMazeGamut-Easy, a smaller $M$ leads to worse test performance, while training on a fixed set of 3000 levels yields comparable generalization performance to a policy trained with value-based level replay on the full level distribution. These results confirm that for certain environments, the auto-curriculum discovered by our method leads to more sample-efficient training on a fixed set of levels than even training on the full level distribution.

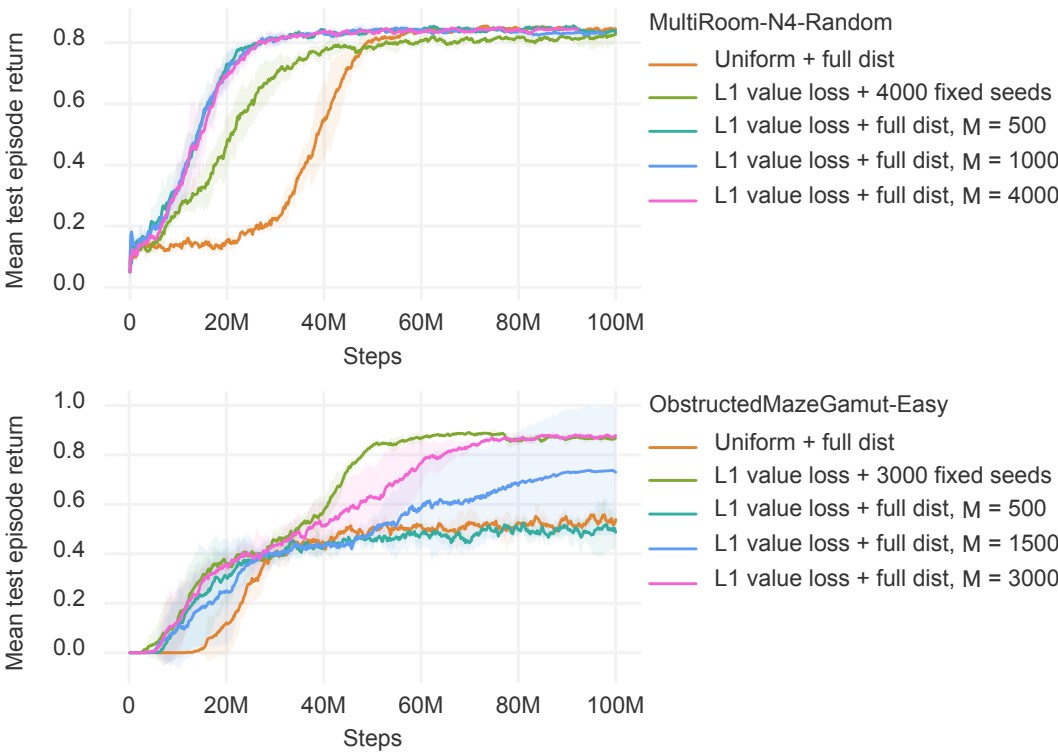

Figure 12: Mean test episodic returns on MultiRoom-N4-Random (top) and ObstructedMazeGamut-Easy (bottom) with access to the full level distribution at training. We set $P_D$ to a Bernoulli parameterized as $p = 0.5$ for MultiRoom-N4-Random and $p = 0.95$ for ObstructedMazeGamut-Easy (the best values found via grid search). As with all MiniGrid experiments using Prioritized Level Replay, we set $\beta = 0.1$ and $\rho = 0.3$.

## C.3 Additive Gains with UCB-DrAC

UCB-AutoDrAC, introduced in Raileanu et al. (2020), has been shown to improve the generalization of PPO agents on Procgen Benchmark by making use of a data augmentation strategy. This method extends PPO by introducing a UCB bandit trained concurrently with the agent. Given a set of predefined image transformations, $\mathcal{F}$, this bandit learns to select, at the start of each PPO update, a single transformation to apply to every frame in the update. Following the standard UCB algorithm, the selection rule for data augmentation $f_t$ before the $t$-th update is defined as

$$f_t = \arg\max_{f \in \mathcal{F}} \left[ Q(f) + c\sqrt{\frac{\log(t)}{N(f)}} \right],$$

where $Q(f)$ is computed as a sliding window of the most recent $K$ mean returns obtained by the agent after the policy was updated on frames transformed by $f$, the constant $c$ is the UCB exploration constant, and $N(f)$ counts how many times transformation $f$ has been selected. Note that the UCB-AutoDrAC algorithm also introduces an additional KL-divergence term in the loss function to ensure the policy exhibits transformation-invariant optimality. We refer the reader to Raileanu et al. (2020) for the full description of this method.

We use the best hyperparameters for Procgen Benchmark reported in Raileanu et al. (2020), setting $K = 10$ and $c = 0.1$. For Prioritized Level Replay, we make use of the L1-value-loss scoring function with $\beta = \rho = 0.1$. Figure 13 shows test performance throughout training of UCB-AutoDrAC with L1-value-loss level replay against UCB-DrAC without level replay and PPO with uniform sampling.

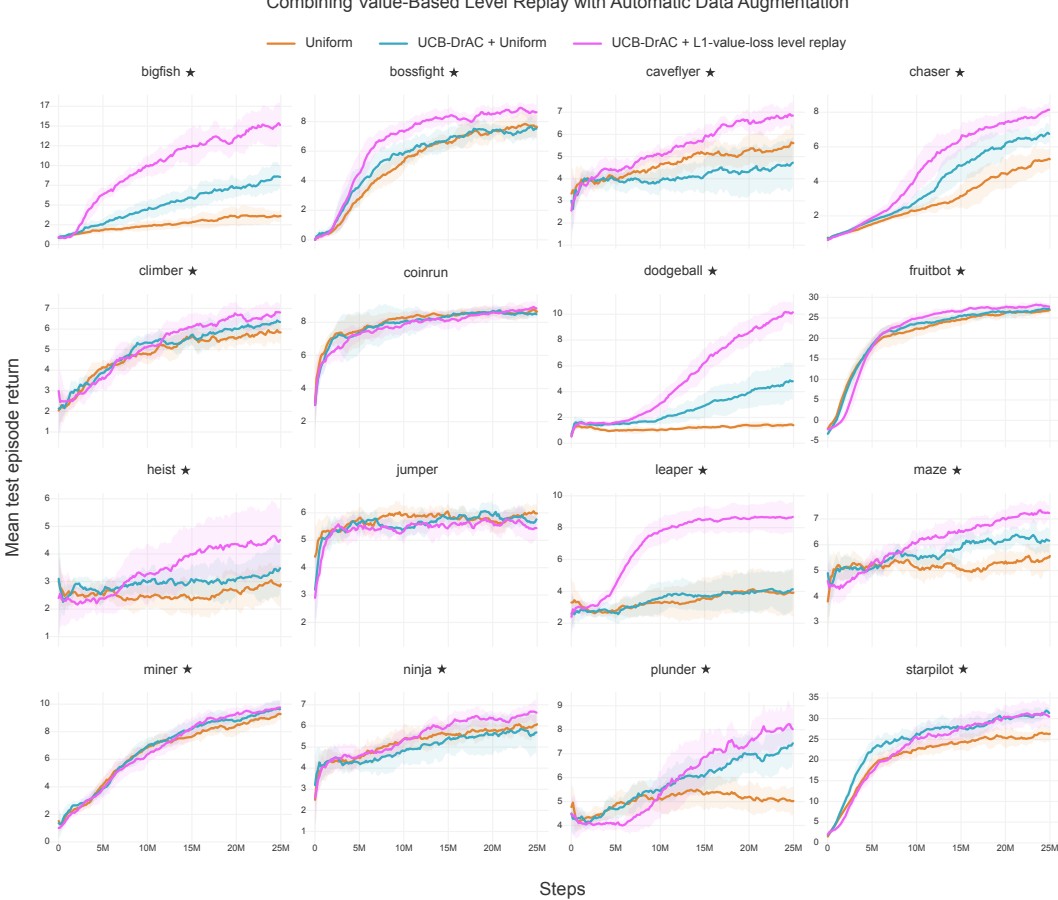

Figure 13: Mean test episodic returns of UCB-DrAC with value-based level replay on Procgen Benchmark (easy), attaining statistically-significant (★) generalization gains on 14 of 16 games.

# D   ALGORITHMS

In this section, we provide detailed pseudocode for how Prioritized Level Replay can be used for experience collection with a policy-gradient method based on $T$-step rollouts. Algorithm 3 presents the extension of the generic policy-gradient training loop presented in Algorithm 1 to the case of $T$-step rollouts, and Algorithm 4 presents an implementation of experience collection in this setting (extending Algorithm 2). Note that when using $T$-step rollouts in the training loop, rollouts may start and end between episode boundaries. To compute level scores on full trajectories segmented across rollouts, we compute scores of partial episodes according to Equation 2, and record these partial scores alongside the partial episode step count in a separate buffer $\tilde{S}$. The function $S$ then technically takes $\tilde{S}$ as an additional argument to combine the partial score with the score computed on the final trajectory segment.

---

**Algorithm 3** Generic $T$-step policy-gradient training loop with prioritized level replay

---

**Input:** Training levels $\Lambda_{\text{train}}$ of an environment, policy $\pi_\theta$, rollout length $T$, number of updates $N_u$, batch size $N_b$, policy update function $\mathcal{U}(\mathcal{B}, \theta) \rightarrow \theta'$.

Initialize level scores $S$, partial level scores $\tilde{S}$, and level timestamps $C$
Initialize global episode count $c \leftarrow 0$
Initialize set of visited levels $\Lambda_{\text{seen}} = \varnothing$
Initialize experience buffer $\mathcal{B} = \varnothing$
Initialize $N_b$ parallel environment instances $E$ to a random level in $\in \Lambda_{\text{train}}$
**for** $u = 1$ **to** $N_u$ **do**
    $\mathcal{B} \leftarrow \varnothing$
    **for** $k = 1$ **to** $N_b$ **do**
        $\mathcal{B} \leftarrow \mathcal{B} \cup$ **collect_experiences**$(k, E, \Lambda_{\text{train}}, \Lambda_{\text{seen}}, \pi_\theta, T, S, \tilde{S}, C, c)$ $\triangleright$ Using Algorithm 4
    **end for**
    $\theta \leftarrow \mathcal{U}(\mathcal{B}, \theta)$
**end for**

---

---

**Algorithm 4** Collect $T$-step rollouts with prioritized level replay

---

**Input:** Actor index $k$, batch environments $E$, training levels $\Lambda_{\text{train}}$, visited levels $\Lambda_{\text{seen}}$, current level $l$, policy $\pi_\theta$, rollout length $T$, scoring function **score**, level scores $S$, partial scores $\tilde{S}$, staleness values $C$, and global episode count $c$.

**Output:** Experience buffer $\mathcal{B}$

  Initialize $\mathcal{B} = \varnothing$, and set current level $l = E_k$
  Observe current state $s_0$, termination flag $d_0$
  **if** $d_0$ **then**
    Choose current level $l_i \leftarrow$ SAMPLENEXTLEVEL($\Lambda_{\text{train}}$, $S$, $C$, $c$) and $E_k \leftarrow l$
    Update level timestamp $C_i \leftarrow c$
    Observe initial state $s_0$
  **end if**
  Choose $a_0 \sim \pi_\theta(\cdot|s_0)$
  t = 1
  Initialize episodic trajectory buffer $\tau = \varnothing$
  **while** $t < T$ **do**
    Observe $(s_t, r_t, d_t)$
    $\mathcal{B} \leftarrow \mathcal{B} \cup (s_{t-1}, a_{t-1}, s_t, r_t, d_t, \log \pi_\theta(a))$
    $\tau \leftarrow \tau \cup (s_{t-1}, a_{t-1}, s_t, r_t, d_t, \log \pi_\theta(a))$
    **if** $d_t$ **then**
      Update level score $S_i \leftarrow \textbf{score}(\tau, \pi_\theta, \tilde{S}_i)$ and partial score $\tilde{S}_i \leftarrow 0$
      $\tau \leftarrow \varnothing$
      Update current level $l \leftarrow$ SAMPLENEXTLEVEL($\Lambda_{\text{train}}$, $S$, $C$, $c$) and $E_k \leftarrow l$
      Update level timestamp $C_i \leftarrow c$
    **end if**
    Choose $a_{t+1} \sim \pi_\theta(\cdot|s_t)$
    $t \leftarrow t + 1$
  **end while**
  **if** not $d_t$ **then**
    $\tilde{S}_i \leftarrow (S(\tau), |\tau|)$                         $\triangleright$ Track partial score and $|\tau|$ for time-averaged scores.
  **end if**

  **function** SAMPLENEXTLEVEL($\Lambda_{\text{train}}$, $S$, $C$, $c$)
    $c \leftarrow c + 1$
    Sample replay decision $d \sim P_D(d)$
    **if** $d = 0$ **and** $|\Lambda_{\text{train}} \setminus \Lambda_{\text{seen}}| > 0$ **then**           $\triangleright$ Sample an unseen level, if any
      Define new index $i \leftarrow |S| + 1$
      Sample $l_i \sim P_{\text{new}}(l|\Lambda_{\text{train}}, \Lambda_{\text{seen}})$
      Add $l_i$ to $\Lambda_{\text{seen}}$, add initial value 0 to $S$ and $C$
    **else**                                                   $\triangleright$ Sample a level for replay
      Sample $l_i \sim (1 - \rho) \cdot P_S(l|\Lambda_{\text{seen}}, S) + \rho \cdot P_C(l|\Lambda_{\text{seen}}, C, c)$
    **end if**
  **end function**

---

