# OpenReview forum: "Prioritized Level Replay"
_ICLR.cc/2021/Conference — Reject_

### Official Review · AnonReviewer2 · 2020-10-18
**Official Blind Review #2**

**Rating:** 6
**Confidence:** 4

**Review:**

##########################################################################

**Summary**:

This paper proposes a prioritized sampling strategy for task sampling in procedurally generated environments. While training an RL agent across many tasks (levels), we can either sample a new task uniformly from the training task distribution or sample a new task with different weights. The paper claims that sampling based on the average magnitude of generalized advantage estimate (GAE) yields faster learning in most Procgen environments and a few MiniGrid environments. Overall, I found the idea to be simple and intuitive. But the benefit of using prioritized level replay is also not very consistent across different environments used in the paper.
##########################################################################

**Strengths**:

The method of the paper is simple and can be incorporated into many existing RL algorithms.

The paper shows that L1 value loss is a good scoring metric for the prioritization by comparing several different choices.


##########################################################################

**Weaknesses**:

The advantage of using prioritized level replay against uniform sampling is rather small in many tasks (11 out of 19 tasks) shown in the paper (Climber, Coinrun, Dodgeball, Fruitbot, Heist, Jumper, Maze, Miner, Ninja, Starpilot, ObstructedMazeGamut-Medium).


The paper only presents results in the easy mode of procgen. While I understand the reason due to the limit on the computational resources, it would be more convincing to show the results on at least 1 or 2 procgen tasks in the difficult mode. If the overall task difficulty is increased, then the advantage of learning in a curriculum (starting from the easy tasks and then to the difficult tasks) are expected to be more salient.


While the scoring metrics used in the paper are all related to the policy function or value function that is being learned, how about a scoring metric that is only based on the number of steps that the agent experiences in a task and whether the agent fails or succeeds? Intuitively, if the lifetime of an agent is short and the agent solves the task, it is an easy task. If the agent does not solve the task or it takes the agent many more steps to solve the task, it is a difficult task. Another metric to compare to is prioritize based on the return value of the trajectories. If the return value is high, then the task is probably already solved by the current policy, so we can sample such tasks less frequently.

In Figure 4, it seems the advantage of using L1 value loss for the prioritization in sampling is more obvious in easy environments (Multiroom-N4-Random and ObstructedMazeGamut-Easy). But its performance becomes very close to the uniform sampling strategy in harder environments (ObstructedMazeGamut-Medium). Why would the advantage of using prioritization (hence implicit curriculum) fade as the task difficulty increases?

In Figure 4, it is hard to connect the top row to the bottom row as the top row uses the environment steps for the x-axis, the bottom row uses the number of PPO updates for the y-axis. I would suggest plot the bottom row figures in terms of the environment steps as well and use the same x-range.


##########################################################################

**Minor points**:

Some details about the experiment setup, especially the MiniGrid environments, are missing. For example, how do the MiniGrid environments look like, what does the difficulty mean in these environments, which parts of the environments are randomized across levels, reward structure, etc.

---

> ### Author Response · Authors · 2020-11-15
> **Responses to your questions and comments (Part 1)**
>
> We thank the reviewer for their feedback that will improve the paper.
> ### On the benefits of Level Replay
> The reviewer states “benefit of using prioritized level replay is also not very consistent across different environments” / “ advantage of using prioritized level replay against uniform sampling is rather small in many tasks”. As stated in our [joint response to all reviewers](https://openreview.net/forum?id=NfZ6g2OmXEk&noteId=f5jXzTsAGK8), we want to strongly emphasize that in the initial submission of the paper we reported *statistically significantly* better generalization (as determined by Welch’s t-test) on the majority (11 out of the 16) of Procgen envs and are on par with the others. On MiniGrid, we report statistically significant improvements in sample efficiency on all 3 environments, and statistically significant gains in final test performance on ObstructedMazeGamut-Easy and ObstructedMazeGamut-Medium. In the updated version of the paper, we now set a new SOTA on Procgen Benchmark when our method is combined with the previous SOTA-method UCB-DrAC, and by a fair margin. In particular, these additional results show its applicability to other methods than standard PPO, setting a new SOTA for the OpenAI Procgen Benchmark via augmenting UCB-DrAC [Raileanu _et al._ 2020](https://arxiv.org/pdf/2006.12862.pdf) with Prioritized Level Replay. This result also concretely demonstrates that our method can enable complementary improvements in combination with other powerful methods for improving generalization, which is a strong advantage of our method—it can be implemented in conjunction with most any other method, and as we see, can thereby provide additive gains in sample-efficiency and generalization performance.
>
> ### Evaluating against OpenAI Procgen Hard
> Using OpenAI Procgen Benchmark “easy” for our experiments is not only based on the high computational demand of training on “hard”, but also due to the precedent in the literature to use “easy” instead of “hard”. For example, see [Laskin _et al._ (NeurIPS 2020) “Reinforcement Learning with Augmented Data”](https://arxiv.org/abs/2004.14990) and [Raileanu _et al._ (2020) “Automatic data augmentation for generalization in deep reinforcement learning.”](https://arxiv.org/abs/2006.12862)
>
> Nonetheless, we recognise the value of at least running some experiments on a selection of hard tasks. We have run experiments on Procgen benchmark’s hard difficulty setting and added them to the paper in Appendix C. We find that Prioritized Level Replay works on these environments as well, specifically resulting in an average gain across games of 39% over PPO with uniform level sampling. This further strengthens our claims about the robustness and applicability of our method. Thank you for suggesting this, and we hope that these additional results give you the confidence you need to fully support the publication of this paper.
> ### Regarding your step-count-based scoring function suggestion
> Thank you for suggesting this additional baseline. We have discussed this a lot over the last few days. Our concerns are:
> 1. That trajectory length is not a signal that meaningfully correlates to difficulty. In some games, e.g. MiniGrid, shorter trajectories mean a better policy (higher return), while on others, e.g. most Procgen Benchmark games, longer trajectories correlate with better policy (surviving longer and therefore obtaining a higher return). This makes step-count-based scoring functions not generally transferable across environments, unlike the TD-error-based scoring functions, which we empirically show work across over a dozen environments.
> 2. Sampling based on return also does not make sense. If you bias toward sampling for high return, you will likely oversample the easy levels, which the agent will master quickly and reinforce this sampling bias, thereby only rarely having a chance at learning on harder levels. If you bias towards sampling low return levels, you will tend to sample hard levels that the agent cannot solve, and the agent will make slower learning progress.
>
> As a result, we are unsure exactly what scoring function would make sense for conditioning on trajectory length and episode success or return. Do you have a specific one in mind? If so, we are happy to attempt to try and run it on some environments and share any results before the end of the discussion period.
>
> That said, we will be open-sourcing the code and hope people will feel empowered to try their own scoring metrics and variants on our ideas. As described in the paper, Prioritized Level Replay describes a general framework for a class of selective-sampling algorithms for sampling training levels in an RL setting. The aim of our paper is to present this framework in addition to empirical studies demonstrating the effectiveness of a specific instance of this class of algorithms, value-based level replay (where the scoring function is the L1 value-loss), across a wide variety of environments.

---

> ### Author Response · Authors · 2020-11-15
> **Responses to your questions and comments (Part 2)**
>
> ### Additional Details about MiniGrid
> These are available in the original MiniGrid paper, but we understand you are suggesting that we include them here to make the paper more self-contained. We agree this would be a useful addition to the paper, and we have added this content to the Appendix in order to better guide people wishing to replicate our setting. Note that we will also be open sourcing all experiment code, which should make these environment details as clear as possible.
> ### Summary
> Thank you again for your feedback and questions. As you will see from our response above, this has guided us in providing additional experimental results which confirm our findings that our method, Prioritized Level Replay, is robust and applicable in a promising range of problems. We hope you will reconsider your assessment in light of this, and stand ready to respond to any further questions or requests you may have to ensure the paper is as strong as it can be.

---

### Official Review · AnonReviewer1 · 2020-10-26

**Rating:** 7
**Confidence:** 3

**Review:**



This paper concerns about the use of experience replay in a way that past experience is sampled based on (implicit) levels so as for the agent to better adapt to the current task at hand. The authors defined a replay distribution (where experience is sampled) based on two scores relevant to learning potential and staleness. Due to its formulation, the change of replay distribution can be used as an outer-layer of a learning algorithm without any modification of the underlying learning mode. The authors conducted experiments over a set of benchmark data sets relevant to level-ness and found statistically significant improvements over more than half of the tasks.

The overall impression of the paper is that it presents a simple yet effective solution to prioritizing experience in the presence of level-ness in a given task. The basic idea is finding out past experience with high "learning potential" by examining a past trajectory's 'wrongness' and how long the policy was not updated (= likely still wrong).

Point: The notion of level and its relevance to learning potential.
First, the paper does not contain any mathematical (or clear) definition of level, which should be crucial to understand the paper. At the beginning it is only explained as different configurations (i.e., any non-singleton environment). Further, it is hard to understand why the notion of levels is even needed to be employed in the paper. An RL agent has a specific way to learn experience (updating parameters) and its artifacts makes "experience replay" useful in most RL agents. Then, there would be an optimal way of replaying experience at any given time --- a certain order of a subset of past trajectories to be replayed for a current policy. The current form of P_replay (Eq. 1) does not need any specific notion of level-ness but only 'learning potential'. I suspect that Eq. 1 also works for a singleton environment, which the authors excluded from consideration.

Point: The conjecture about curriculum learning.
It is reasonable to assume that the notion of hardness of a task for an RL agent is the difficulty of optimizing its policy (=resulting in higher TD-errors). When the human understanding of easiness of a task (i.e., level) matches the agent's ability to optimize, we would safely say that PLR induces curriculum implicitly. It is nice to see such plots (Figure 4) that empirically validate the conjecture. However, isn't it a much anticipated result?

Questions
Q1. Would different algorithms other than (PPO + GAE) make the results different from the current form?

Q2: If we interpret P_S and P_C as two probability distributions, multiplying them seems more natural to me. What is the rationale behind for adding them not multiplying them (or use (1-rho) log P_S + (rho) log P_C)? Further, any reason for P_C being proportional to c-C_i?

Q2. How about learning hyper-parameters on the fly? Both \beta and \rho might be adjusted throughout learning.
Further, it is conceivable that the optimal \beta and \rho are not fixed quantities but can be dependent to a given pair of policy and trajectory.

Minor
Figure 1, there are two taus. The top would be \pi?
Background "We to refer to"
=======
I read through all the reviews and rebuttals and I could better understand and evaluate the paper. I updated my score to 7.

Given that this replay scheme works fairly well (intuitively, empirically), easy to understand and implement, fairly sufficient amount of empirical experimentation, I would like to see the paper accepted (and adopted and improved by others).

One more comment about staleness.
I think staleness is a proxy measure for the (unmeasured) score of the 'current' policy on that level. So I would like to see (in future or revised version) some experiments that measure how well staleness measure correlate with such score. Further, the way staleness is designed properly reflects how the score degrades as the level isn't played.

Some idea.
It would be nice to make a connection to multi-task learning where tasks share some similarities. Currently, level is somewhat 'linearly' defined. If an agent plays level x, then staleness for level x' (something similar to x') doesn't have to be updated a lot compared to another task which might be dissimilar to level x. Hence, some similarity measure can be further employed (or learn a metric).

---

> ### Author Response · Authors · 2020-11-15
> **Responses to your questions and comments (Part 1)**
>
> We thank the reviewer for their feedback to improve our paper. From reading your review, we are not very clear as to why the score is low given the fairly positive tenor of the review. We are confident that the points you make are addressed both in the revisions to the paper made based on your feedback, and by the responses below. We hope this alleviates any concerns you might have, and that you will be prepared to support the paper or further explain what stands between the paper and a supportive assessment on your part.
>
> ### On the notion of replay
> We believe the reviewer has misunderstood fundamental aspects of our method. As defined in the paper, we use “replay” to refer to sampling a *new* trajectory from a level, *not* training on past trajectories collected from that level, e.g. from a replay buffer—which is not performed by standard policy-gradient methods such as PPO used in this paper. We appreciate this distinction is only clearly drawn in page 2 of the paper, and we have tweaked the abstract and introduction to improve clarity on this point.
> ### On combining the staleness and score distributions
> The reviewer asks **“if we interpret P_S and P_C as two probability distributions, multiplying them seems more natural to me. What is the rationale behind for adding them not multiplying them”**. Could the reviewer please clarify why taking the product seems more natural? That would be a mechanism for taking the joint probability of two random variables. Here we have two distributions over one random variable (which level to sample), and thus we induce a mixture over them by taking the convex combination (which yields, of course, a valid and normalized distribution), as is done in mixture models such as GMMs. There is no canonical or “one true way” of doing such combinations of distributions, but we believe this is as close to standard as it comes in statistics.
> ### On the definition of level
> We intentionally make a weak assumption of what a level is as this method is generally applicable to any environment in which variations of the environment instances can be determined by some index value, e.g., a seed, a named environment configuration, etc. We are unsure what would be gained from an attempt to formally pin this down, but are happy to hear from you regarding this. Alternatively, would you be satisfied with further examples of what might constitute a “level”, at the point in the paper where the term is introduced?
> ### On surprisingness
> The reviewer states **“the improvement in empirical results is not particularly surprising”**. We respectfully strongly disagree: it is not obvious at all, as only a single score function worked well, and there are intuitive reasons to believe each of the tested score functions should work. For example, we imagine the reviewer would agree with us that using policy entropy is a perfectly valid hypothesis for inducing a curriculum that leads to improved generalization—however our experiments show that the opposite is the case.
>
> Moreover, as discussed in Section 5.1 and further shown in Appendix C, the method only provides gains when the score-based distribution is mixed with the staleness-based distribution, while sampling from either of these two distributions separately does not work. There is a fairly intuitive explanation for this, post-hoc, which is that scores drift increasingly “off-policy” as the staleness increases, and the mixture cancels this out. We will include a brief mention of this in our results section, but we maintain that this is a novel and un-intuitive finding. Fortunately, most findings become intuitive and unsurprising once explained.

---

> ### Author Response · Authors · 2020-11-15
> **Responses to your questions and comments (Part 2)**
>
> ### Learning hyperparameters on the fly
> Like most methods, ours also introduces a couple of new hyperparameters that can be meta-learned. While completely correct, this is a suggestion that can be applied to any RL method, in any setting, that does not perform on-the-fly meta-learning. Methods like PBT (e.g. [ROMUL](https://openreview.net/forum?id=9MdLwggYa02)) can be used to do this, but application of such a method seems quite orthogonal to the goals of the method presented in this paper. Further, our experiments show that our method is fairly robust to hyperparameter decisions and the same set of hyperparameters were shared successfully across all games in Procgen, and a separate set shared across all MiniGrid environments—so while meta-learning the hyperparameters on-the-fly may improve our results, we did not find doing so necessary to showcase the effectiveness of our method.
> ### Clarifying Figure 1
> The reviewer asks **“Minor Figure 1, there are two taus. The top would be π?”**. This is not a mistake: a decision is made every time new experience is solicited by the training process — do we play a new unseen level or a previously seen one (and in either case, gathering *new* experience from the level we end up choosing to play)? This decision can be described as two mutually exclusive “paths” to obtaining the next training trajectory tau. The two taus in Figure 1 belong to these independent paths, and the second argument of the score function is always the policy in its current state (see Equation 2 of Section 3.1). Thank you for suggesting that we could make this Figure clearer and more self-contained by discussing this in the caption. We have improved the caption accordingly.
>
> ### Summary
> We hope to have addressed each of your concerns to your satisfaction, and made corresponding improvements to the paper which make you more comfortable supporting its publication. If you have any further questions or comments, we are happy to receive them during this discussion period.

---

### Official Review · AnonReviewer4 · 2020-10-28
**Well done paper, but unclear significance / potentially limited applicability**

**Rating:** 5
**Confidence:** 3

**Review:**

### Paper Summary

This paper allows agents to set the initial conditions (level) for procedurally generated episodes during exploration to past observed values, and proposes to have agents form an intrinsic curriculum by resampling past levels based on a heuristic measure of expected learning progress. The authors test several heuristic measures and find that the average absolute magnitude of the generalized advantage estimate works well. The authors hypothesize that this intrinsic curriculum will improve optimization/learning relative to an agent that always samples initial conditions from the environment distribution. The authors verify that their prioritization strategy usually improves performance in several Progen Benchmark and MiniGrid environments, usually by a small but statistically significant amount, but sometimes by a large amount.

### Summary Review (highlights re: quality, clarity, originality and significance)

The paper is well written and clear after one understands the basic idea. The idea is simple, and the algorithm/experiments seem straightforward to reimplement. The experiments are about what one would expect and seem to be well executed. The idea is original but not particularly innovative (this seems like the first heuristic prioritization approach that would come to mind given that the agent is able to choose the level). The improvement in empirical results is not particularly surprising (if anything, I would have expected more large improvements like the ones on bigfish/leaper environments). As this method is constrained to procedurally generated environments (or at least, evaluation of the method is constrained to procedurally generated environments), the significance seems rather limited. The required assumption seems rather strong, as it requires a simulator / control over the environment, which limits applicability.

### Pros

- This a simple idea that can improve performance in Procedurally Generated Environments given that the agent is allowed to set the initial conditions / pick the level.
- The performance improvement in 4 of the 19 environments tested is large & seems absolute (i.e., it's seems like a final performance improvement, not just a sample efficiency improvement).
- The paper is well written/presented, easy to understand, and the empirical evaluation seems well done. The results do not seem difficult to replicate.

### Cons

- Despite being less intrusive than direct access to the level generation mechanism, the assumption that the agent can replay levels seems rather strong to me, and simplifies the task of learning procedurally generated environments very substantially.

    ($\dagger$) I would argue that we don’t use procedurally generated environments as benchmarks in order to solve procedurally generated environments, but rather a tool for measuring generalization, so it's unclear to me that a technique that improves sample efficiency only in a procedurally generated environment is useful.

    Unlike environment-agnostic techniques like prioritized replay, HER, intrinsic reward, intrinsic goal selection, etc., this requires you to have control over the environment, which seems to limit the applicability. If this is only useful with a simulator, then the small gains in sample efficiency aren’t actually that relevant, though this approach does seem to improve final performance in 4 of the 19 environments tested.
- It’s not clear until the second page whether your method is a prioritized replay buffer scheme, or a task selection scheme. Actually, I was certain it was a prioritized replay buffer scheme until the second page, because that is the more natural/general setting (as noted above, I find the assumption that the agent can replay levels to be rather strong).
- Several new hyperparameters are introduced; this said, guidance/ablations are performed, and it seems like the choices will generalize decently well (albeit there were different choices for ProcGen/Minigrid).

### Questions / Etc.

- My main question for the authors is to ask for a counterargument to ($\dagger$) above.
- It would be good if this can be shown to work in multi-goal setting, as it is quite similar to ProcGen setting... you draw some distinctions, but I do think your approach would be applicable there.

---

> ### Author Response · Authors · 2020-11-15
> **Responses to your questions and comments**
>
> We thank the reviewer for their feedback that we will use to improve our paper.
> ### On the applicability of this method
> The reviewer states: **“evaluation of the method is constrained to procedurally generated environments”/“unclear to me that a technique that improves sample efficiency only in a procedurally generated environment is useful”**. As we explain in our [joint response to all reviewers](https://openreview.net/forum?id=NfZ6g2OmXEk&noteId=f5jXzTsAGK8), using procedurally generated environments for evaluation is not a weakness but instead a main strength of our approach compared to work that relies on specific properties of an underlying environment instance, e.g. Atari games in the Arcade Learning Environment. Procedurally generated environments are much harder to master due their constant stream of novel observations that the agent has to generalize towards. Many previously successful methods (e.g. Go-Explore, count-based exploration, etc. to name just a few) would fail as they assume the environment stays fixed and the agent can memorize trajectories to high value regions in the state space.
>
> The reviewer further states: **“it requires a simulator / control over the environment, which limits applicability”:**
> Note that assuming one can replay a level (or, more concretely, any configuration of the environment) is a much weaker assumption than assuming there is only a *single* configuration the agent needs to do well in (as it is the case in Atari). In contrast to Atari, we test for systematic generalization by sampling completely unseen seeds (and thus environment instances) at test time.
>
> Additionally, note that any environment that displays random initialized states can be viewed as procedurally-generated. Take for example, a robot reaching task that starts with a random arrangement of objects. In this case, “resetting the seed” during training would equate to simply returning the objects to the corresponding initial arrangement, and a completely manageable task for real-world training, which requires resetting the initial arrangement of objects at the start of each episode anyway. If we are instead training in simulation with access to the simulator (which is more often than not in RL), then why not take advantage of it? As the goal of training on procedurally-generated environments is to test for and improve generalization, if a training strategy uses a privileged action—such as resetting the env seed at the start of training episodes—does not subvert the integrity of the test time evaluation protocol of the agent, then we should by all means take advantage of such a strategy.
> ### Regarding sample efficiency and performance
> The reviewer also states **“small gains in sample efficiency aren’t actually that relevant, though this approach does seem to improve final performance in 4 of the 19 environments tested”**:
> As stated in our [joint response to all reviewers](https://openreview.net/forum?id=NfZ6g2OmXEk&noteId=f5jXzTsAGK8), we strongly emphasize that in the initial submission of the paper we reported *statistically significantly* better generalization (as determined by Welch’s t-test over 10 runs) on not 4 but **11** out of the 16 Procgen Benchmark envs, 3 out of 3 MiniGrid environments tested (in terms of sample efficiency and/or final test returns), and are on par for the others. In the updated version of the paper, we now report a new SOTA on Procgen benchmark when our method is used in combination with a UCB-DrAC agent, reporting statistically significant gains on 14 of 16 games, with much higher gains on average per game.
>
> ### On defining the notion of replay
> Thank you for bringing to our attention that the notion of replay (as in gathering *new* experience from a level) used in our paper, in contrast to the notion of replay used in experience replay, is not clear until the second page. We agree, and we have changed the writing of the abstract and introduction to reflect this. We trust you will find the paper is improved as a result, but please let us know if it is somehow still in need of additional clarity.
> ### Summary
> Thank you for your pertinent questions and comments. We hope the responses, and the improvements we have made to the paper in response to your feedback, have convinced you that the paper is worthy of your support. We strongly believe it proves the concept and is rigorously evaluated and compared, including (as of this revision) against the state of the art on the OpenAI Procgen Benchmark (which it improves upon). Naturally, there are further experiments to be done, including investigating application of this method to looser notions of “level” outside of procgen (e.g. starting states or, as you suggest, multi-goal settings), but these constitute ambitious and exciting matter for future work we hope to investigate. In the meantime, we would be grateful for your support for this paper, and are happy to further discuss outstanding concerns you may have, if any.

---

### Official Review · AnonReviewer3 · 2020-11-03
**Review [Updated]**

**Rating:** 7
**Confidence:** 3

**Review:**

**SUMMARY**

The present work considers the problem of learning in procedurally generated environments. This is a class of simulation environments in which each individual environment is created algorithmically where certain environmental factors are varied in each instance (referred to as levels in this work). Learning algorithms in this setting typically use a fixed set of training and evaluation environments. The present work proposes to sample the training environments such that the learning progress of the agent is optimized. This is achieved by proposing an algorithm for level prioritization during training. The performance of the approach is demonstrated on the Procgen Benchmark and two MiniGrid benchmarks and the authors argue that their approach induces an implicit curriculum in sparse reward settings.

**STRENGTHS**
- The general idea of prioritization for level sampling makes a lot of sense and is demonstrated to improve sample-efficiency for skill learning in procedurally generated environments.
- I also liked that the authors compared with a big variety of different scoring metrics.

**WEAKNESSES**
- The intuition of "greater discrepancy between expected and actual
returns, making $\delta_t$ a useful measure of the learning potential" makes sense. The heuristic score also works well in practice. One limitation I see is that there is no theoretical justification for why the TD-error is a good predictor for learnability.
- This is maybe more an avenue for future work than an actual weakness but it seems to me that the algorithm is not making use of all potentially useful information. In each timestep, it only considers the last score achieved in a level. Maybe it would also be interesting to consider the full history of scores. My intuition is that levels in which agents were historically very slow to learn are maybe not as useful (or at least not useful at the moment). I.e., maybe in order to learn competing at such levels it is better to compete on other levels first?
- Is there, at least from a qualitative perspective, an explanation for why certain environments do not benefit as much from the proposed level sampling approach?

**REPRODUCIBILITY**

The work seems reproducible. Most of the information relevant for reproducibility is given in Appendices A & B. It would be great if the authors would also make the source code available.

**CLARITY**

Overall, I found the work to be very clearly written and have only minor questions/remarks:
- To what extent does the use of TD-errors potentially limit the type of learning algorithms that can be used in the context of the proposed framework. Computing the TD-error requires a value function. As I understand it, some RL algorithms never compute a value function.
- If I haven't overlooked it, there is no explanation of $c$ after eq. (4) while $C_i$ is explained earlier. Is $c$ simply the current episode?


**EVALUATIONS**

The work is compared with several scoring function baselines using PPO. While the authors claim that the method is applicable to other RL agents, the evaluations do not show any results with other agent types. The authors mention several different benchmarks in that space. It would be interesting to know why particularly Procgen Benchmark and MiniGrid environments were chosen.

It is also not clearm to me why PPO is used as the base agent. Was this for ease of implementation / its popularity? Wouldn't it make sense to use more recent agents to see the added benefit of the proposed approach. E.g., would V-MPO be applicable here?

**NOVELTY / RELEVANCE**

The work is very interesting and the authors make a compelling case that procedurally generated environments can benefit from a conscious sampling of the levels with regard to usefulness for learnability.

I am not sure whether the claim "Prioritized Level Replay induces an implicit curriculum, taking the agent gradually from easier to harder levels." is fully valid. As I understand it, the hardest levels are also the most likely to be sampled. The force counteracting this to some extent is the staleness-based sampling term $P_C$. For a gradual curriculum, I would expect $P_S$ to be designed such that it does not choose the hardest level but the one promising the best learning outcome. Particularly in the early stages of the training, the hard levels might be less useful than levels of medium difficulty.

**SUMMARY**

I found that paper very interesting. While I am not working in the particular subfield of the work and cannot sufficiently judge relation with prior works, I can confidently say that the idea and implementation details were conveyed very well. My main concerns are regarding the understanding of the "failure cases" and to what extent the graduality claim applies. That being said, I believe this line of work to be really interesting and to have a lot of potential for improved sample-efficiency when training RL agents in algorithmically generated simulation environments.

**POST-DISCUSSION UPDATE**

I want to thank the authors for correcting my misunderstandings, answering my questions, and providing additional material. As a consequence of this, I have raised my score to "Accept". To answer your question about what would be needed for a higher score: For a strong accept recommendation, I would have expected a mix of several additional things such as a clear impact outside of own subfield, code availability at time of submission (to evaluate how easy it is to reproduce the results and re-use the code), or more additional theoretical justification (in the sense of new formal guarantees for at least certain aspects of the proposed method). While not directly working in this subfield, I still think this work is solid and worthy of publication.

---

> ### Author Response · Authors · 2020-11-15
> **Responses to your questions and comments (Part 1)**
>
> We thank the reviewer for their insightful and detailed comments that will improve our paper. It is great to hear the reviewer found our approach sensible, our writing clear, and that they praised the large variety of scoring functions that are explored. We address, here and in our [post summarizing the changes to the paper](https://openreview.net/forum?id=NfZ6g2OmXEk&noteId=f5jXzTsAGK8), your main concerns, and hope that this will lead to you considering strengthening your recommendation or explaining what still stands in the way, so that we may further improve the paper.
> ### Clarification about curriculum “As I understand it, the hardest levels are also the most likely to be sampled”
> We respectfully believe this is a misconception. At the end of Section 5.1 on page 7, we state “easier levels result in non-zero, non-stationary returns earlier in training, while harder levels give rise to near stationary returns until the agent learns an improved policy that allows making further progress on the level [...] sampling levels according to the L1 value-loss then leads to an implicit curriculum from easier to harder levels.” What matters for a score of a level to be high based on L1 value-loss is whether or not the agent, given the current policy, over- or under-estimates value, which does not consistently correlate with the relative difficulty of the environment. For instance, in early stages of training the agent might correctly estimate low value for harder levels, thus sampling more frequently easier levels to improve it’s policy before gradually sampling harder levels more often. This is a hypothesis that we verify qualitatively in Figure 4.
>
> ### On the limits imposed by the use of TD error
> In our work, we investigated policy gradient methods, which near-uniformly make use of a value estimate. This does not cover all approaches to RL, but a large class of state-of-the-art actor-critic policy-gradient methods such as: PPO, IMPALA, A2C/A3C, A2C-AKTR, APPO, and Phasic Policy Gradients.
>
> We agree it would be interesting to investigate whether the core mechanisms of Prioritized Level Replay can also be combined with value-based RL methods (e.g. DQN) for future research. However, this is outside of the scope of the present paper, and is something we are investigating in follow-up work.
>
> ### Theoretical justification of TD error for scoring learning potential:
> As motivated in the paper, the TD error is the difference between the empirical return and the predicted return. When this discrepancy is high, there is a greater opportunity for the agent to learn. In fact, the same reasoning is used in Schaul _et al._ 2016 (Prioritized Experience Replay) to motivate the use of TD errors as a learning signal for ranking the utility of sampling *past* transitions in the experience replay buffer.
>
> Further, the advantage-based gradient estimator used in nearly all actor-critic methods, including PPO which is used in our paper, entails computing nearly the same TD-error terms, so our use of TD-error-based scores may be seen as roughly correlating with the value of the gradient estimate resulting from the last trajectory taken over each level.

---

> ### Author Response · Authors · 2020-11-15
> **Responses to your questions and comments (Part 2)**
>
> ### On why certain environments do not benefit as much
> We believe it is generally an important open question in the field of RL as to why certain methods work well for some environments while not others. This question is especially relevant to procedurally-generated environments, in which we could ideally arrive at a theory that predicts which factors of variation within the environment exist and which RL methods can best handle them. In short, we feel this is an important question, though one whose proper answer should be left for separate, future work.
>
> That said, one observation we make is that the environments for which Prioritized Level Replay does not significantly improve performance seem to fall into two categories: 1. Environments on which the policy does not significantly improve for most of training under standard PPO with uniform level sampling (e.g. heist and miner). At a high level, these may simply be games that are inherently challenging for policy-gradient methods to learn on, regardless of the order of the training levels and more strongly require forms of generalization that cannot be improved through curriculum learning, requiring methods such as data augmentation instead. 2. Environments on which other methods also only make small or negligible improvements to sample-efficiency (e.g. maze and jumper). For this latter category, it is possible standard PPO can already quickly learn a policy effective at generalizing, so additional methods like Prioritized Level Replay and DrAC provide only marginal benefit.
>
> ### On evaluation concerns
> We used PPO because it was the method employed in OpenAI’s original paper on Procgen ([Cobbe _et al._ 2019](https://arxiv.org/abs/1912.01588)), as well as nearly all follow-up work tackling this benchmark [Igl _et al._. 2019](https://arxiv.org/abs/1910.12911), [Lee _et al._ 2020](https://arxiv.org/abs/1910.05396), [Laskin _et al._ 2020](https://arxiv.org/abs/2004.14990), [Raileanu _et al._ 2020](https://arxiv.org/abs/2006.12862), and [Igl _et al._ 2020](https://arxiv.org/abs/2006.05826)). The well-established base of prior work using PPO provides an established set of benchmark performance results and makes it an ideal base actor-critic algorithm on which to evaluate gains resulting specifically from our method for selectively sampling training levels. What matters most in our evaluation is to show that the selective sampling employed by Prioritized Level Replay results in improved generalization and sample efficiency compared to not using our method. Using a nonstandard RL algorithm on Procgen Benchmark could lead to even more performance gains, but these would only be incidental and unrelated to showing the benefit of using Prioritized Level Replay.
>
> Further, the original Procgen paper also shows that PPO outperforms Rainbow, an off-policy method based on DQN, which further motivates our focus on actor-critic policy-gradient methods for Procgen. Lastly, another reason we favor PPO is that it is a widely used algorithm that is relatively easy to implement and fast.
>
> Regarding trying in combination with different RL agents, as outlined in our post about changes to the paper, we ran our method in combination with a UCB-AutoDrAC agent—the previous SOTA on Procgen Benchmark, and are happy to share that using our method in combination with UCB-AutoDrAC leads to a new state-of-the-art on Procgen benchmark. Thank you for suggesting this improvement to our evaluations. We hope that this demonstration of the complementarity of our method with recent improvements in this area, which forms as new SOTA for ProcGen, sufficiently addresses your suggestion to the point you would consider strengthening your recommendation
>
> ### Regarding the value of $c$: "If I haven't overlooked it, there is no explanation of  $c$ after eq. (4) while $C_i$ is explained earlier. Is  simply the current episode?"
> Correct. As stated in Section 3.2 of our paper, "Here, $c$ is the count of total episodes sampled so far in training."
>
> ### Summary
> Given that your review is overall very positive, and we believe we have addressed all of your concerns with extended results and clarifications, we would be interested in hearing from you what is still missing for a strong acceptance of the paper. We, of course, are fully committed to a complete release of the source code to enable others to adapt our method to their settings. In fact, the latest codebase is largely already prepared for public release.

---

### Author Response · Authors · 2020-11-15
**Joint response to reviewers (Part 1)**

We thank the reviewers for their time and their feedback. We are confident that in responding to individual questions and comments, the paper has already improved significantly. We have prepared a major update of the paper with new state-of-the-art results. We believe this update addresses all of the major points raised by you. Below we summarize the main improvements and additional results that we have added to our revision. In addition, we would like to address misunderstandings that concerned multiple reviewers, which we expand upon in a more tailored form in individual responses.

### UPDATE: New State-of-the-Art Results
We are pleased to share new results empirically demonstrating that our method combines easily with the previous state-of-the-art method, UCB-DrAC, to set a new state-of-the-art (in terms of generalization performance) on the OpenAI Procgen Benchmark. These results will be included in our rebuttal revision, and we include an updated table listing test performance of the various methods we investigated here:

|Games|Uniform|UCB-DrAC|PLR (Ours)|UCB-DrAC + PLR (Ours)|
|------|------|------|------|------|
|bigfish|3.72&pm;1.24|8.73&pm;1.13|10.91&pm;2.81|**14.28&pm;2.12**|
|bossfight|7.7&pm;0.37|7.65&pm;0.67|**8.94&pm;0.35**|8.84&pm;0.8|
|caveflyer|5.37&pm;0.79|4.61&pm;0.93|6.32&pm;0.47|**6.76&pm;0.7**|
|chaser|5.23&pm;0.69|6.79&pm;0.93|6.9&pm;1.21|**8.01&pm;0.6**|
|climber|5.93&pm;0.6|6.39&pm;0.92|6.33&pm;0.84|**6.8&pm;0.67**|
|coinrun|8.62&pm;0.4|8.62&pm;0.45|8.76&pm;0.47|**8.95&pm;0.37**|
|dodgeball|1.69&pm;0.23|5.11&pm;1.65|1.78&pm;0.46|**10.33&pm;1.36**|
|fruitbot|27.29&pm;0.94|27.02&pm;1.35|**28.02&pm;1.35**|27.62&pm;1.47|
|heist|2.77&pm;0.92|3.17&pm;0.74|2.93&pm;0.48|**4.93&pm;1.3**|
|jumper|5.71&pm;0.42|5.61&pm;0.46|5.83&pm;0.48|**5.86&pm;0.34**|
|leaper|4.18&pm;1.33|4.44&pm;1.42|6.83&pm;1.15|**8.66&pm;0.98**|
|maze|5.46&pm;0.37|6.21&pm;0.5|5.49&pm;0.8|**7.23&pm;0.82**|
|miner|8.73&pm;0.72|**10.09&pm;0.6**|9.56&pm;0.62|10.03&pm;0.54|
|ninja|6.04&pm;0.41|5.83&pm;0.79|**7.24&pm;0.38**|6.96&pm;0.5|
|plunder|5.05&pm;0.55|7.79&pm;0.89|**8.68&pm;2.18**|7.67&pm;0.95|
|starpilot|26.84&pm;1.54|**31.68&pm;2.36**|27.9&pm;4.35|29.64&pm;2.22|
|PPO-Normalized (%)|100.0&pm;4.5|129.77&pm;8.18|128.29&pm;5.83|**176.4&pm;6.12**|

As summarized in this table, our method, Prioritized Level Replay, matches the previous state-of-the-art method, UCB-DrAC when used in isolation, while yielding an average 76.4% improvement over PPO when combined with UCB-DrAC. Additionally, the improvements of this combined approach over just using UCB-DrAC by itself are statistically significant.

Here, normalized performance is computed as done in the UCB-AutoDrAC paper (Raileanu _et al._ 2020): For each method and game, we evaluate the final policy on 100 held-out test levels across 10 training seeds. We divide the average score per game for each method by the corresponding average game score attained by PPO and aggregate these normalized scores across all 16 games and report the mean and standard deviation across the 10 seeds.

---

> ### Author Response · Authors · 2020-11-15
> **Joint response to reviewers (Part 2)**
>
> ### Training on Procgen Benchmark’s hard setting (*R2*):
> We thank R2 for their suggestion to also assess the effectiveness of our method on the hard difficulty setting of Procgen Benchmark. We share the initial results below in table form, showing that similar to the easy difficulty setting, Prioritized Level Replay improves test performance on the hard difficulty setting. We believe this result further establishes the robustness of our method and provides yet more compelling evidence that the order of environment instances in which we train agents will impact the sample-efficiency and generalization performance of the final policy, and that further, such an improved ordering can be discovered automatically on-the-fly over the course of training using our method. We will include the results in the update to our paper and are appreciative of R2 for suggesting this useful addition to our paper.
>
> |Games|Uniform|PLR (Ours)|
> |------|------|------|
> |bigfish|**9.13&pm;4.51**|7.77&pm;1.01|
> |bossfight|6.82&pm;0.59|**8.67&pm;0.72**|
> |caveflyer|3.13&pm;0.47|**6.36&pm;0.08**|
> |chaser|5.3&pm;1.22|**6.26&pm;0.67**|
> |climber|3.26&pm;0.46|**6.23&pm;0.76**|
> |coinrun|5.06&pm;0.24|**5.42&pm;0.39**|
> |dodgeball|1.76&pm;0.29|**2.01&pm;1.09**|
> |fruitbot|11.15&pm;2.59|**15.86&pm;1.26**|
> |heist|0.84&pm;0.36|**1.24&pm;0.37**|
> |jumper|3.3&pm;0.46|**3.58&pm;0.46**|
> |leaper|2.52&pm;1.45|**6.42&pm;0.43**|
> |maze|3.98&pm;0.18|**4.12&pm;0.46**|
> |miner|9.5&pm;0.15|**9.67&pm;0.44**|
> |ninja|3.14&pm;0.33|**5.36&pm;0.52**|
> |plunder|2.72&pm;0.34|**4.1&pm;1.32**|
> |starpilot|**2.85&pm;0.7**|2.63&pm;0.3|
> |PPO-Normalized (%)|100.0&pm;9.47|**138.64&pm;9.62**|
>
> ### Training on the full level distribution:
> While the experiments in our paper focus on the standard Procgen Benchmark generalization evaluation protocol of training on a fixed budget of levels and testing on the full level distribution, we included additional experimental results of using Prioritized Level Replay to train on the full level distribution on MiniGrid environments in Appendix C. To do this, we use a modified version of Prioritized Level Replay that keeps a running level buffer of information (i.e. seeds, scores, and timestamps) of up to only the top M levels with the highest learning potential at any point in training for updating the level replay distribution. Our results show that when both methods are given access to the full level distribution at training, sampling levels via Prioritized Level Replay still outperforms uniform level sampling. This result further solidifies the flexibility and robustness of our method, demonstrating it can also provide statistically significant improvements to generalization performance when training on the full level distribution.

---

> ### Author Response · Authors · 2020-11-15
> **Joint response to reviewers (Part 3)**
>
> ### Significantly improved generalization (*R2* & *R4*):
> In our initial submissions, Prioritized Level Replay demonstrates improved generalization compared to the standard practice of uniform level sampling in 11 out of 16 environments, not just 4 out of 16. These improvements are statistically significant as determined by Welch’s t-test. We emphasize here that the reported improvements are measured in terms of *test* performance, not training performance. Furthermore, we also verify in our revision that the improvements in sample-efficiency on the 3 MiniGrid environments are also statistically significant, and the improvements in final test performance on both ObstructedMazeGamut-Easy and ObstructedMazeGamut-Medium are also statistically significant. Further, our results indicate that Prioritized Level Replay used by itself without any auxiliary methods for improving generalization performance already matches the previous SOTA method, UCB-AutoDrAC, while it sets a new SOTA on Procgen Benchmark when used in combination with UCB-AutoDrAC (see above). The improvements of this combined approach over both PPO and UCB-AutoDrAC used by itself are statistically significant.
>
> ### “Limitation” to procedurally generated environments (*R4* & *R3*):
> We chose procedurally generated environments like OpenAI Procgen Benchmark and MiniGrid as they are established benchmarks for testing systematic generalization of RL and led to a large number of publications on exploration (e.g.  [Goyal _et al._ 2019](https://arxiv.org/abs/1901.10902),  [Raileanu and Rocktäschel 2019](https://arxiv.org/abs/2002.12292), [Campero _et al._, 2020 (AmiGo)](https://arxiv.org/abs/2006.12122)), meta-learning (e.g. [Alet _et al._ 2019](https://arxiv.org/abs/2003.05325), [Co-Reyes _et al._ 2019](https://arxiv.org/abs/1811.07882)), and generalization (e.g. [Igl _et al._ 2019](https://arxiv.org/pdf/1910.12911.pdf), [Igl _et al._ 2020](https://arxiv.org/abs/2006.05826), [Laskin _et al._ 2019](https://arxiv.org/abs/2004.14990), [Raileanu _et al._ 2020](https://arxiv.org/abs/2006.12862)), to name just a few. In contrast to other benchmarks like Atari, we, as well as a growing body of researchers (e.g. [Cobbe _et al._ 2019 (Procgen Benchmark](https://arxiv.org/abs/1812.02341)), [Risi and Togelius 2020 (Nature paper arguing for why PCG environments are useful for furthering generality of ML methods)](https://www.nature.com/articles/s42256-020-0208-z?proof=t), and [Küttler _et al._ 2020 (NetHack Learning Environment)](https://arxiv.org/abs/2006.13760) all argue that procedurally generated environments are important for evaluating RL methods as they pose challenges not present in previous benchmarks. In fact, recently successful approaches to exploration (e.g. [Go-Explore](https://arxiv.org/abs/1901.10995)) rely heavily on the determinism and the low size of the observation space in the respective non-procedurally generated MDP (Atari’s Montezuma’s Revenge). Thus, we see it as a core strength and contribution that our method is capable of successfully utilizing procedurally generated levels to reach a new state-of-the-art in this challenging class of RL problems.

---

> ### Author Response · Authors · 2020-11-16
> **Quick update: Paper now updated to the latest version**
>
> We would like to inform the reviewers and AC that our submission has now been updated with the latest version of our paper: https://openreview.net/pdf?id=NfZ6g2OmXEk.

---

### Decision · Program_Chairs · 2021-01-07
**Final Decision**

**Decision:**

Reject

**Comment:**

The paper presents a method for automatically generating levels of varying complexity for training the agent.  The results are well summarized in the paper abstract, "significantly improved sample-efficiency and generalization on the majority of Procgen Benchmark environments as well as two challenging MiniGrid environments." The work is clearly presented, and the experiments are thorough.

R1, R2, and R3 voted to accept the paper with 7, 6, and 7 scores. R4 voted to reject the paper with a score of 5. The reviewers mostly agree (except for R4) that significant performance gains have been achieved. R4 is unsatisfied as he/she believes that performance gains are small and exploit the simulator (e.g., using resets).

The paper's main pro is well summarized by R4's comment, "The method of the paper is simple and can be incorporated into many existing RL algorithms."

The main drawback of the paper is that many curriculum learning techniques have been proposed in the past. E.g., Matiisen et al. (https://arxiv.org/pdf/1707.00183.pdf). In fact the authors discuss this work in the related work section, but dub it multi-agent RL work. This is not true. The method of Matiisen et al. is very similar to the proposed approach but uses a different criterion for learning progress. Comparison to this work is warranted, without which the paper should not be accepted. In the post-rebuttal discussion, R2 and R3 agree that this comparison is necessary. Therefore, I recommend that this paper be rejected for now and resubmitted to a future venue after incorporating a comparison with Matiisen et al.